

# A multiplex centrality metric for complex social networks: sex, social status, and family structure predict multiplex centrality in rhesus macaques

Brianne Beisner[1,2], Niklas Braun[3], Márton Pósfai[4,5], Jessica Vandeleest[2], Raissa D'Souza[3,4,5] and Brenda McCowan[1,2]

[1] Department of Population Health & Reproduction, School of Veterinary Medicine, University of California, Davis, Davis, CA, United States of America

[2] Neuroscience and Behavior Unit, California National Primate Research Center, Davis, CA, United States of America

[3] Department of Mechanical and Aerospace Engineering, University of California, Davis, Davis, CA, United States of America

[4] Department of Computer Sciences, University of California, Davis, Davis, CA, United States of America

[5] Complexity Sciences Center, University of California, Davis, Davis, CA, United States of America

Corresponding author
Brianne Beisner,
babeisner@ucdavis.edu

## ABSTRACT

Members of a society interact using a variety of social behaviors, giving rise to a multi-faceted and complex social life. For the study of animal behavior, quantifying this complexity is critical for understanding the impact of social life on animals' health and fitness. Multilayer network approaches, where each interaction type represents a different layer of the social network, have the potential to better capture this complexity than single layer approaches. Calculating individuals' centrality within a multilayer social network can reveal keystone individuals and more fully characterize social roles. However, existing measures of multilayer centrality do not account for differences in the dynamics and functionality across interaction layers. Here we validate a new method for quantifying multiplex centrality called consensus ranking by applying this method to multiple social groups of a well-studied nonhuman primate, the rhesus macaque. Consensus ranking can suitably handle the complexities of animal social life, such as networks with different properties (sparse vs. dense) and biological meanings (competitive vs. affiliative interactions). We examined whether individuals' attributes or socio-demographic factors (sex, age, dominance rank and certainty, matriline size, rearing history) were associated with multiplex centrality. Social networks were constructed for five interaction layers (i.e., aggression, status signaling, conflict policing, grooming and huddling) for seven social groups. Consensus ranks were calculated across these five layers and analyzed with respect to individual attributes and socio-demographic factors. Generalized linear mixed models showed that consensus ranking detected known social patterns in rhesus macaques, showing that multiplex centrality was greater in high-ranking males with high certainty of rank and females from the largest families. In addition, consensus ranks also showed that females from very small families and mother-reared (compared to nursery-reared) individuals were more central, showing that consideration of multiple social domains revealed individuals whose social centrality and importance might otherwise have been missed.

## INTRODUCTION

Social life is complex and multi-faceted. Members of a society interact with each other in a variety of different ways and these different behaviors are often interdependent. In animal societies, for example, individuals may spend time in proximity, groom one another, and direct aggression at or show submission to one another. Social relationships thus arise from the patterning of these different types of interactions, and similarly, social structure arises from the patterning of these relationships within a society (*Hinde, 1976*; *Whitehead & Dufault, 1999a*). Traditional analytical approaches cannot adequately represent or quantify such multilayer complexity. Yet, given the importance of social relationships for the fitness, health and well-being of social animals (e.g., *Akinyi et al., 2013*; *Silk et al., 2009*; *Silk et al., 2010*; *Cameron, Setsaas & Linklater, 2009*; *Stanton & Mann, 2012*), accurate quantification and representation of this type of social complexity is critical to advance our understanding of the selective forces acting on individuals, the impact individuals may exert on their conspecifics, and the overall social structure of animal societies. Here we present a new approach for quantifying multilayer network centrality, called 'consensus ranking' (*Pósfai et al., 2019*), and illustrate its utility for uncovering novel information about individuals' social roles and social structure using rhesus macaques (*Macaca mulatta*).

### Social network analysis for social complexity

Animal societies are complex systems; they are composed of multiple individuals that interact in a variety of contexts using multiple behaviors which generates higher-order emergent properties (e.g., hierarchies; social structure) (*Bradbury & Vehrencamp, 2014*; *Fisher & Pruitt, 2019*). The recent surge in the application of social network analysis (SNA) to study animal social relationships and social structure reflects this recognition (*Flack et al., 2006*; *Sundaresan et al., 2006*; *Wey et al., 2008*; *Ramos-Fernández et al., 2009*; *Beisner et al., 2011a*; *McCowan et al., 2011*; *Pinter-Wollman et al., 2011*; *Brent et al., 2013*; *Pasquaretta et al., 2014*; *Ilany, Booms & Holekamp, 2015*). SNA is well-suited for examining social complexity as it offers a way to mathematically represent not just direct relationships but indirect ties among conspecifics, thus quantifying the web of indirect social connections in which members of society are embedded.

Most empirical work examining social networks has focused on single network layers. Certainly, analyses of single network layers have offered valuable insights. For instance, network analysis has revealed examples of the social diffusion of behavior (*Aplin et al., 2012*; *Allen et al., 2013*), the impact of individual centrality and network modularity on infection risk and transmission (*Griffin & Nunn, 2012*; *VanderWaal et al., 2013*; *VanderWaal et al., 2014*; *Balasubramaniam et al., 2016*), key aspects of species social structure (*Lusseau, 2003*; *Sundaresan et al., 2006*; *Pinter-Wollman et al., 2011*; *Beisner et al., 2016*), and the impact of key individuals on social structure (*Lusseau & Newman, 2004*; *Flack et al., 2006*). Yet, these
social patterns and processes are more complex than what can be represented by a single network.

## Multilayer networks

A multilayer network is composed of two or more different networks (for example, each representing a different type of interaction) that may be linked to each other through inter-layer edges (*De Domenico et al., 2013*; *Bianconi, 2018*). A multiplex network is a special type of multilayer network and inter-layer edges connect the same actor in different layers, which generally equates to different layers representing different sets of behavioral interactions among the same set of individuals. Given that members of a society interact in different ways using different social behaviors, each interaction type can represent a different layer of a multiplex social network. Multilayer network approaches unify information across separate interaction networks or social domains (*Barrett, Henzi & Lusseau, 2012*; *Finn et al., 2019*) to address questions about both individual-level (e.g., individual social role) and higher-order (e.g., group-level social structure) social patterns. For instance, using a single network layer to identify the individual(s) of greatest influence, importance, and/or centrality runs the risk of missing truly influential individuals by relying on incomplete social information. Similarly, higher-order social structure results from the interplay among multiple different social domains, e.g., dominance and kinship, and their joint consideration is essential for accurately characterizing species social structure (e.g., *Thierry, 2004*). In fact, a recent investigation of what causes social instability in captive social groups of macaques found that socially stable groups were all characterized by the same pattern of interdependence between the aggression network and the status signaling network, whereas the groups that succumbed to social collapse exhibited deviations from this pattern (*Beisner et al., 2015*), indicating that social stability emerged from a complex relationship between multiple network layers.

Much evidence already points to multilayer social effects on key outcomes related to fitness and health. For instance, the composite sociality index (CSI), which is calculated using both proximity and grooming relationships, is associated with greater infant survival (*Silk et al., 2009*) and longevity (*Silk et al., 2010*) in female baboons. Infection risk and transmission of bacterial infectious agents has been linked to multiple behavioral networks in both wild meerkats and captive macaques, with both tuberculosis and *Shigella* infection risk, respectively, being jointly associated with position in a dominance network and an affiliation network (*Drewe, 2010*; *Balasubramaniam et al., 2016*). In addition, fecal *E. coli* transmission in captive macaques is jointly associated with grooming and huddling network centrality (*Balasubramaniam et al., 2019*). However, examining behavioral networks separately fails to acknowledge the complex interdependence between different interaction layers, and composite measures such as CSI do not factor in indirect social connections like many network measures do. Furthermore, aggregating multiple types of interactions into a single weighted network does not preserve layer-specific topology and dynamics or interdependencies across interaction layers which may be important for understanding higher-order social structure or the influence of specific individuals (*De Domenico et al., 2013*; *Halu et al., 2013*). Multilayer network techniques offer a different way to look at

'composite' measures of social behavior that explicitly incorporate interdependence across different types of interactions, indirect social connections, and the unique dynamics of each contributing layer. Thus, the impact of complex social processes on individual fitness may be better understood and characterized by using multilayer approaches such as multiplex centrality.

Multiplex centrality can further give insights into an individual's social position/ role within society. Individuals are heterogeneous in their social role, impact on others, and relative importance to the maintenance of social structure. Some individuals, for instance, can play a disproportionately important role (or have disproportionately large impact on the lives of others), thus occupying a keystone position in the society (*Modlmeier et al., 2014*). However, a single network layer is unlikely to fully represent the nature or complexity of this role. For example, both alpha males and matriarchs are considered keystone individuals in some societies, and their importance stems from multiple sources. Alpha males may police intra-group conflict (*Bernstein & Sharpe, 1966*; *Flack, De Waal & Krakauer, 2005*; *McCowan et al., 2011*; *Von Rohr et al., 2012*), receive a disproportionate amount of subordination signals thereby affecting their social power (*Flack & Krakauer, 2006*; *Beisner et al., 2016*), be popular grooming partners (*Sade, 1972a*; *Jack & Fedigan, 2018*), or be the most spatially central animals in the group (*Jack & Fedigan, 2018*). Similarly, matriarchs may possess knowledge of key resources or potential threats (*McComb et al., 2001*; *McComb et al., 2011*; *Brent et al., 2015*), contribute to survival of grandchildren (*Hawkes, O'Connell & Blurton Jones, 1997*; *Kachel, Premo & Hublin, 2011*), or uphold the stability of their family's social status (*Beisner et al., 2011a*; *Wooddell et al., 2016*). Given the complexity that can underlie keystone status, multilayer approaches for understanding centrality may better characterize the layers contributing to an individual's keystone status and may identify keystone individuals that would otherwise have been missed.

## Multilayer approaches

Methods to perform multilayer network analysis have only recently been developed, and their application to understanding animal social relationships and social structure has only just begun (*Barrett, Henzi & Lusseau, 2012*; *Chan et al., 2013*; *Beisner et al., 2015*; *Silk et al., 2018*; *Finn et al., 2019*). Some approaches focus on network-level patterns such as characterizing interdependence between network layers (*Barrett, Henzi & Lusseau, 2012*; *Chan et al., 2013*; *Beisner et al., 2015*) and others focus on individual or node-level position such as centrality or clustering (*Barrett, Henzi & Lusseau, 2012*; *De Domenico et al., 2013*; *Smith-Aguilar et al., 2018*). Here we focus on individual-level centrality in a multilayer network.

In network terminology, centrality measures the extent to which a particular individual (i.e., node) is central versus peripheral in the network. Centrality in single and multilayer networks can be assessed using a variety of different metrics, including degree (one's direct connections), betweenness (the tendency to connect individuals who are otherwise weakly connected), eigenvector centrality (the extent to which one is connected to well-connected others), or other node-level measures of connectivity. Centrality in a multiplex network broadly refers to individuals or nodes that are well-connected not simply in a

single layer, but well-connected across multiple network layers. Methods for quantifying centrality in a multiplex network range from simple (e.g., aggregation of connections across layers regardless of interaction type) to complex (encode connections within and between layers using a multi-dimensional tensor) (e.g., *Barrett, Henzi & Lusseau, 2012*; *De Domenico et al., 2015*). For instance, 'Multiplex PageRank' is a method which works by calculating PageRank centrality on one layer and then biasing the PageRank on the next layer by allowing the centrality of a node in the first layer to be influenced by its centrality in the second layer (*Halu et al., 2013*). The open-source software *MuxViz* allows you to calculate multilayer generalizations of multiple different types of centrality such as degree and eigenvector centrality (*De Domenico, Porter & Arenas, 2015*), and between-layer connections are represented by placing an edge between each node and itself across all pairs of layers (*Kivelä et al., 2014*; *De Domenico et al., 2015*).

State-of-the-art measures of multiplex centrality such as versatility and multiplex PageRank (*Bianconi, 2018*; *De Domenico et al., 2015*; *Halu et al., 2013*) assume that all layers represent similar types of relationships and have similar structure, and some are restricted in practice to only two layers (e.g., multiplex PageRank). For instance, an airline transportation network, in which each layer represents a different airline company, is a common example of a multilayer network in which the meaning of a link in each layer is the same (i.e., a flight between two cities in the network) and the network structure is similar (e.g., each airline company has one or more 'hub' cities which operate more flights than the other cities in the network). Yet animal social networks are frequently composed of layers with significant structural differences (e.g., dominance network links are directed whereas affiliative network links may not be; proximity networks are dense, with many of the total possible links being present, whereas contact interaction networks are much more sparse, with far fewer links present; (*Smith-Aguilar et al., 2018*)). Such structural differences across layers may mean that, for example, eigenvector centrality is best for measuring centrality in a dominance interaction layer with clear 'hubs' but betweenness centrality is ideal for measuring centrality in an affiliation layer with evidence of familial/kin clustering. No existing method for quantifying multilayer centrality simultaneously satisfies all requirements: (i) allowing different metrics of centrality on different layers (degree centrality on one layer, betweeness centrality on another), (ii) allowing significant structural differences between layers (directed-undirected, sparse-dense), and (iii) including more than two layers. Here we apply a new method for measuring multiplex centrality, called consensus ranking (*Pósfai et al., 2019*), to multiple social groups of a well-studied nonhuman primate, the rhesus macaque, to examine whether attributes of individuals (sex, age, dominance rank and certainty) or socio-demographic factors (matriline size, mother- vs. nursery-rearing history) are predictive of their overall centrality or influence in the social network. We also illustrate the utility of this method for uncovering novel information about social roles and social structure by comparing the findings from the analysis of multilayer consensus rankings to the analyses of single network layers.

## Rhesus macaque societies as a testbed for multilayer model development

An important step in developing and validating a new multilayer network metric is to identify a sufficiently well characterized and complex model testbed. Rhesus macaques (*Macaca mulatta*) are just such a model species—their behavior and social structure are both complex and well-studied. Rhesus macaque social networks have been studied with regard to social style (*Sueur et al., 2011*; *Balasubramaniam et al., 2018*), genetic variance (*Brent et al., 2013*), health outcomes (*Vandeleest et al., 2016*), and group stability (*McCowan et al., 2011*; *Beisner et al., 2015*; *Beisner et al., 2016*). Furthermore, macaque social structure is thought to arise from specific combinations of aggression intensity, dominance asymmetry, and rates of post-conflict affiliation (*Thierry, 2004*), an idea that invokes multilayer social structure.

## Study system and predictions

The existing knowledge of rhesus macaques is rich enough to generate clear predictions about how attributes and socio-demographic factors influence multiplex centrality. Rhesus macaque societies have clear dominance hierarchies and strong kinship relationships among females (*Thierry, 2004*). Sex differences in rank acquisition patterns shape the roles that males and females play in multiple interaction networks. Male-biased dispersal and female philopatry (*Drickamer & Vessey, 1973*; *Sugiyama, 1976*) result in temporally stable, heritable dominance relationships among females (*Sade, 1969*) and more labile dominance relationships among males (*Berard, 1999*). Since dominant animals are expected to be central in single interaction layers heavily influenced by dominance rank (e.g., aggression, submission), we predicted they would also be central in the multiplex network. Furthermore, the effect of dominance on multiplex centrality may differ by age-sex class because high-ranking adult males often play keystone roles—they frequently police the conflicts of group members (*Bernstein & Sharpe, 1966*; *Beisner & McCowan, 2013*) and such policers receive a disproportionate number of subordination signals (*Beisner et al., 2016*). Specifically, we predict high-ranking males and older males will be more central in the multiplex network than lower-ranking or younger males.

Another aspect of dominance that may influence multiplex centrality is dominance certainty. Dominance certainty reflects the average level of certainty (or ambiguity) in an individual's dominance relationships and is based upon pathway consistency in aggression networks (e.g., *Vandeleest et al., 2016*). Low dominance certainty reflects the potential for rank mobility (i.e., gain or lose rank) and may impact multiplex centrality, particularly for males because rank is more labile. For example, a dominant male at risk of losing his rank may be an unpopular grooming partner and he may receive few subordination signals and police conflicts infrequently. We predict that high-ranking males with high dominance certainty will have greater centrality in the multiplex network than high-ranking males with low dominance certainty.

Third, family structure is likely to influence centrality as the presence of kin shapes the patterning of social interactions because the benefits of cooperating with kin are typically greater than with non-kin (*Hamilton, 1964*; *West Eberhard, 1975*). Here, we focus on the

 

impact of family size. Female macaques preferentially affiliate and interact with close kin (*Gouzoules & Gouzoules, 1987*; *Bernstein, Judge & Ruehlmann, 1993*). We predict that females from larger families will have greater multiplex centrality because they have more kin partners with whom to interact than individuals from smaller families.

Finally, we examine rearing history because among captive macaques, nursery-rearing is known to impact brain development and social behavior (*Winslow et al., 2003*; *Suomi, 2006*). Nursery-rearing (NR) refers to individuals that were removed from their mothers during infancy, typically at birth or around weaning age (∼6 months) and housed with same-aged peers. We expect NR to impact social behavior in a cognitively sophisticated social species such as rhesus macaques because much social behavior is learned, initially through an infant's attachment to its mother which influences a wide range of social and physiological outcomes (*Capitanio et al., 2006*). Compared to their mother-reared (MR) counterparts, NR infants explore and play less and exhibit more extreme behavioral and physiological responses to social separation at 6 months of age (*Suomi, 2006*). As they mature, NR monkeys become less affiliative and more aggressive than their MR peers (*Winslow et al., 2003*; *Suomi, 2006*), and spend more time alone (*Winslow et al., 2003*). These behavioral differences have been linked to alterations in neural and endocrine systems (*Clarke, 1993*; *Sánchez et al., 1998*; *Winslow et al., 2003*; *Capitanio et al., 2006*). However, while the evidence to date suggests that NR should impact individuals' abilities to navigate their social environment, most studies of NR vs. MR differences are examined in pair-housed or small group housed animals up to 3 years of age, as opposed to large social groups of adult animals. Of the research that has been done on social groups, results are mixed. Although NR individuals may be lower-ranking than MR individuals when housed together in social groups (e.g., *Bastian et al., 2003*; *Dettmer et al., 2017*), some studies show no discernible differences related to rearing (*Bauer & Baker, 2016*). We predict NR individuals will have lower multiplex centrality than MR.

# MATERIALS & METHODS

## Study subjects

Behavioral network data were collected from seven captive social groups of rhesus macaques (*Macaca mulatta*) housed in half-acre (0.2 ha) outdoor enclosures at the California National Primate Research Center in Davis, CA. Groups varied in size, age structure, number of matrilines, and the rearing history of animals (Table 1). Housing conditions were similar for all groups. Each enclosure contained multiple A-frame structures, suspended barrels, swings, and perches and animals were free to interact as they chose. Animals were fed a standard diet of monkey chow twice per day at approximately 0700 h and between 1430 and 1530 h. Fresh fruit or vegetables were provided once per week and seed mixture was provided daily. Water was available ad libitum. All research reported here adhered to the recommendations in the Guide for the Care and Use of Laboratory Animals of the National Institutes of Health, and the laws of the United States government. This research was approved by the University of California, Davis Institutional Animal Care and Use Committee, protocol #18525.
**Table 1  Study group characteristics.**

| Group | Adult group size (males, females) | Age range[a] yrs (mean) | Matrilines | Matriline size | Percent Nursery-reared[b] | Year studied |
|---|---|---|---|---|---|---|
| A | 87 (30, 57) | 3–19 (7.2) | 26 | 1–17 | 14.9% | 2012 |
| B | 99 (26, 73) | 3–28 (7.5) | 15 | 1–22 | 0 | 2012 |
| C | 101(27, 74) | 3–29 (8.0) | 15 | 1–20 | 0 | 2013 |
| D | 55 (16, 39) | 3–11 (6.0) | 7 | 1–36 | 0 | 2013 |
| E | 101 (34, 67) | 3–11 (5.9) | 37 | 1–15 | 8.9% | 2014 |
| F | 96 (28, 68) | 3–21 (8.3) | 14 | 1–24 | 5.2% | 2014 |
| G | 81(24, 57) | 3–21 (8.2) | 30 | 1–17 | 18.5% | 2016 |

Notes.

[a]Age range of adult subjects measured in years, starting with the youngest subjects at 3 years of age.

[b]Percent of group members that experienced maternal deprivation by being permanently separated from their mother (with subsequent peer- or nursery-rearing) prior to reaching 12 months of age.

Of the total 620 subjects, 42 experienced nursery-rearing. Eighteen subjects were removed from their mothers at birth and housed with one or more peers until the time of group formation. The remaining 24 subjects were mother-reared until 3–11 months of age (mean = 6.9 months) and housed with one or more peers until the time of group formation. Nursery-reared subjects were older, on average, than other subjects (NR subjects: mean = 15.0 years, range = 9–21 years; MR subjects: mean = 6.8 years, range = 3–29 years) because most of them were the founding members during new group formations.

## Behavioral data collection

Groups were observed for six consecutive weeks (except Group G, which was observed for four consecutive weeks) by a team of two observers. Data were collected for 6 h per day on 4 days per week from 0900–1200 h and 1300–1600 h. Previous analyses have confirmed that this frequency of data collection is sufficient to construct reliable networks (*Balasubramaniam et al., 2016*). Groups were part of a larger study on social perturbations and health, and data come from the baseline phase of this study.

Data were collected on both agonistic and affiliative interactions. For agonistic interactions, we used an event sampling design to collect all instances of agonism, including both aggression and status signaling. Agonistic events were recorded as a series of dyadic interactions in which observers documented the identities of all participants, the type of aggressive or submissive behavior used by each, and the sequence of these interactions. Aggressive behavior was categorized according to severity and included threat (e.g., open mouth stare, brow flash), mild aggression (threat and follow, lunge, push, slap, chase <6 m), moderate aggression (grapple, wrestle, chase >6 m), and severe aggression (pin to the ground or bite). In macaques, many agonistic interactions become polyadic (involving more than two participants) when a third-party enters, or intervenes upon, the conflict. Intervention was defined as a third-party entering an on-going conflict by aggressing or approaching one or both participants. More specifically, impartial interventions, which are sometime referred to as policing interventions (*Flack, Krakauer & De Waal, 2005*; *Beisner & McCowan, 2013*), were those in which the intervener treated both participants the same, either by approaching or aggressing them. Status signals were defined as a submissive
response to a peaceful (i.e., no overt aggression) approach and included silent-bared teeth display (SBT), freeze/turn away, or move away (i.e., displacement).

For affiliative interactions, we used a scan sampling design to collect grooming and huddling interactions. Scan samples were conducted every 20 min during data collection hours, proceeding from one side of the enclosure to the other (e.g., from left to right). Grooming was defined as an animal cleaning or manipulating the fur of another individual. Huddling was defined as the occurrence of all forms of body-contact e.g., ventral contact or an embrace between two individuals. A high level of inter-observer reliability was maintained across all observers for all behavioral data collection (Krippendorf's alpha $\geq 0.85$).

## Dominance rank calculation

Dominance ranks were calculated from dyadic aggressive interactions using the *Perc* package in R, a method which combines dominance information from direct interactions and indirect network pathways into the win-loss matrix (*Fushing et al., 2011*; *Fujii et al., 2015*). For instance, a network pathway from A to B to C (i.e., A defeats B, B defeats C) suggests that A is likely dominant to C; the imputed dominance information from network pathways (i.e., each path is a fraction of a win) is added to the win-loss matrix. Since dyadic aggression data were used to calculate both dominance ranks and aggression layer centrality, these dominance ranks could not be used in the analysis of aggression layer centrality (see *Network Layer Construction* and *Statistical Analyses* below). However, study groups were part of a larger study involving social perturbations, and aggression data were available for the experimental phase, which immediately followed the baseline phase and was identical in duration and data collection design. We therefore calculated a second set of dominance ranks from the experimental phase for analyzing aggression layer centrality. We imputed the ranks of animals that were experimentally removed from the group due to the social perturbation ($N = 1 - 19$ animals per group; median = 2 animals) by assuming that their position in the hierarchy was the same as before removal. Dominance rankings calculated from the baseline and experimental phases were highly correlated for all study groups (mean Spearman's $r = 0.967$, range: 0.958–0.976).

## Network layer construction

Five behavioral network layers were chosen to represent multiple key facets of rhesus macaque social structure and social life: (1) dyadic aggression (including all levels of severity), (2) status signaling (i.e., displacements and subordination signals), (3) impartial conflict interventions (aka policing), (4) grooming, and (5) huddling. Layers for dyadic aggression and status signaling were chosen because rhesus macaque society is hierarchically structured. Furthermore, rhesus macaques in this captive population have been found to exhibit a clear relationship between the dyadic aggression and status signaling network layers which, when it erodes or changes significantly, is associated with social collapse (*Beisner et al., 2015*). The impartial conflict policing layer was chosen because impartial interventions, performed most frequently by powerful individuals (e.g., alpha and/or beta males), are important for the control of within group conflict and maintenance of

social stability (*Bernstein & Sharpe, 1966*; *Flack, Krakauer & De Waal, 2005*; *McCowan et al., 2011*). Furthermore, the connections between intervener and combatants may reflect more than differences in social power but also valued social relationships, as there is evidence that policers defend victims (*Beisner & McCowan, 2013*). The layers for grooming and huddling were chosen because of the importance of affiliative interactions for the maintenance of social cohesion (*Lehmann, Korstjens & Dunbar, 2007*).

Each layer differed in its properties. Aggression, status signaling, and impartial policing layers were treated as directed networks because edge direction contains important information about social dominance that yield emergent structures such as the dominance hierarchy and social power distribution (*Beisner & McCowan, 2013*; *Beisner et al., 2016*). The layers for grooming and huddling were treated as undirected networks because links represent the presence of an affiliative relationship (rather than the direction of a given affiliative interaction) and connectivity reflects broad social cohesion (e.g., grooming can be directed up the hierarchy (*Seyfarth, 1977*), toward kin or close allies (*Sade, 1972b*), yet all these interactions facilitate social cohesion).

## Multilayer consensus ranking

To generate consensus rankings, we selected an appropriate centrality metric for each layer. Dominance-based layers (i.e., aggression, status signaling, policing) were measured using centralities based on finding hubs such as eigenvector (*Bonacich, 2007*) and degree centrality, whereas affiliative layers (i.e., grooming, huddling) were measured using betweenness centrality which is based on cohesion. Eigenvector centrality was used in the aggression layer because it is effective at determining hierarchy in connected networks. Degree centrality was chosen for the status signaling (in-degree) and policing (sum of in- and out-degree) layers because these networks were sparser and not always fully connected (Table 2). For the policing layer, both out-going and in-coming edges contained valuable information with regard to power differences and valued social relationships. Betweenness centrality was chosen because it better reflects cohesion in our study system's affiliation networks, which can include functionally different types of interactions in the same layer (e.g., grooming among kin, which can create clustering structure, and 'economic' grooming, which may connect across matrilines). All network layers were weighted, and edge weights were calculated as frequencies of interaction during the study period.

Our multilayer consensus ranking approach combines the centrality information across the different network layers by first generating a ranking of importance for each layer, i.e., in each layer, individuals are ranked 1 to a maximum of $N$, from largest centrality to smallest centrality (where $N$ is the number of nodes). We treat the cases when two or more nodes share the same centrality value by assigning them the same rank. For example, if individuals A, B, and C have centralities 0.4, 0.4, and 0.2, respectively, then A and B are ranked first and C is ranked second. Then, a rank aggregation algorithm is used to find the most central nodes in the multiplex network (*Arrow, 2012*). In general, a rank aggregation method takes two or more rankings of the same items as an input and produces a consensus ranking. The specific definition of consensus ranking depends on the rank

**Table 2** Network density (the proportion of all possible edges that are observed in the network) for all five network layers across all seven study groups.

| Group | Aggression | Status | Police | Groom | Huddle |
|-------|-----------|--------|--------|-------|--------|
| A | 0.271 | 0.085 | 0.021 | 0.259 | 0.190 |
| B | 0.219 | 0.090 | 0.010 | 0.188 | 0.299 |
| C | 0.261 | 0.086 | 0.015 | 0.235 | 0.244 |
| D | 0.355 | 0.177 | 0.018 | 0.346 | 0.314 |
| E | 0.243 | 0.086 | 0.012 | 0.237 | 0.316 |
| F | 0.332 | 0.138 | 0.019 | 0.237 | 0.132 |
| G | 0.182 | 0.120 | 0.010 | 0.212 | 0.211 |

aggregation algorithm of choice. Here, we focus on the widely-used and intuitive Borda count (*De Borda, 1781*; *Dwork et al., 2001*).

Our version of Borda count works by first assigning a score to each node equal to their average rank in the individual layers. More specifically, let $b_{\alpha,i}$ be rank of node $i$ in layer $\alpha$; then the score assigned to node $i$ is

$$b_i = \frac{1}{L} \sum_{\alpha=1}^{L} b_{\alpha,i},$$

where $L$ is the number of layers in the multiplex. Borda count then assigns a consensus rank to each node based on the scores $b_i$: The node with the lowest score is determined to be the highest overall ranked candidate; the node with the second lowest score is the second overall ranked candidate; and so on.

A rank aggregation algorithm is beneficial for quantifying multilayer centrality because (a) it provides a means of combining information across layers with different topologies by allowing different centrality metrics for each layer, and (b) all layers are weighted equally so that no single layer has greater influence than another. Often the actual importance of each layer relative to others is unknown or speculative at best, and the weight assigned to each layer in an aggregate network measure is either arbitrary or based on the size of the network (e.g., edge count, average degree). One potential weakness, however, is that the ranking in each layer is done separately and therefore does not incorporate the multiplexity of interdependence across layers. For instance, an individual connected to the same 10 individuals in layers A and B will be higher ranked by consensus ranking than an individual connected to a different set of six individuals in layers A and B (12 unique individuals overall).

### Rank tiers within layer

The values of a centrality metric are unevenly distributed, typically the centrality difference between high-ranked nodes is large, while the centralities of low-ranked nodes are almost indistinguishable (Fig. 1). This information is lost if nodes are placed in a linear ordering, prompting the question: when do we assign a different rank to two nodes? There is no established answer to this question. Here, instead of assigning an ordinal rank to each node, we place the nodes into tiers using one dimensional hierarchical clustering (*Defays, 1977*;

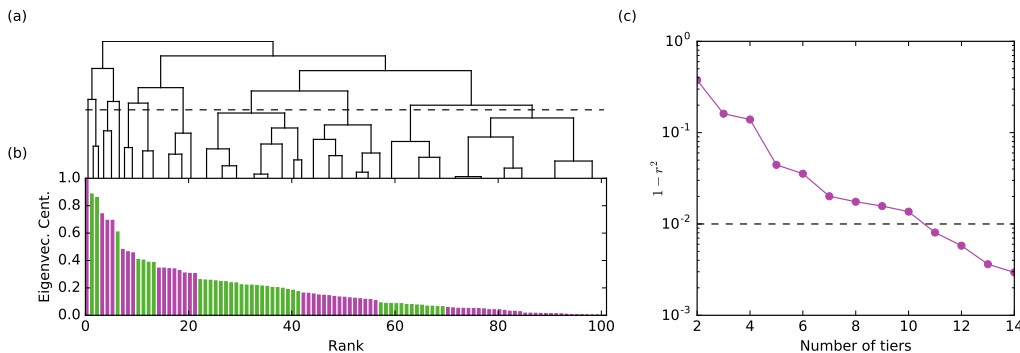

**Figure 1** **The method for assigning rank tiers is illustrated.** Hierarchical clustering was performed on the centrality values for a given network layer. The hierarchical clustering tree shown in (A) maps onto the plot of centrality values ordered from largest to smallest shown in (B) such that all members of the same cluster are assigned the same rank tier (different rank tiers denoted by alternating colors green and purple in the bar graph in (B)). Plot (C) shows $1 - r^2$ (the correlation between the original and tiered centralities) with respect to the number of tiers. The final number of rank tiers selected (i.e., the height at which the clustering tree is cut) is the first one which falls below the threshold line (where 99% of the variance is explained). Illustrated is the aggression layer for group B.

*Müllner, 2011*). At the start of the procedure each node is placed in its own cluster and the distance between each node is the absolute value of their centrality. Then we merge the two closest nodes into a cluster and iteratively repeat this procedure until all nodes are joined. To measure the distance between two clusters with more than one node we use complete linkage, that is their distance is given by the maximum distance between any pair of nodes in the two clusters. This use of complete linkage ensures that we find tiers with a relatively similar spread in values (*Everitt et al., 2011*). The process of tiering distinguishes groups of nodes with similar centrality, while maintaining a ranked structure.

Hierarchical clustering does not produce a single grouping of the nodes but provides a hierarchy of possible clusters represented as a hierarchical tree (Fig. 1A). Cutting the tree at different heights yields tiers at different resolutions. The choice of the cut is somewhat subjective and is typically decided based on data heuristics or the requirements of the application. We chose the location of the cut considering how much of the variance of the un-tiered centrality values is explained by the tiered centrality values (i.e., a 90% threshold means we cut the tree producing the minimum number of tiers such that the explained variance is at least 90%). The explained variance is quantified by $r^2$, where $r$ is the Pearson correlation between the original centralities and their tiered values (all nodes in a tier are assigned the mean centrality of the nodes in that tier). The threshold value affects the number of tiers we identify: for one study group, a 90% threshold produces 4-8 tiers per layer, while a 99% threshold yield 7–12 tiers per layer. We chose a 99% threshold for all layers and groups because (a) it produced enough layers to sufficiently differentiate individual monkeys within a group for the purposes of our analysis (e.g., 100 monkeys spread across 7–12 rank tiers) and (b) it mapped well onto visually-obvious break points in the data. For validation purposes, we also analyzed the data using a 99.5% threshold

to determine if results were robust to number of tiers. Tier identification and Borda rank calculations were run using Python code developed by co-authors NB and MP.

## Statistical analyses

To estimate the extent to which different layers contain similar versus different information about individuals' social position within the social network, we first examined the correlation between each pair of layers for each study group. Pairs of layers with higher Spearman correlations contain more similar information. We then used a recently-developed machine learning clustering method, Data Mechanics (*Fushing & Chen, 2014*; *Guan & Fushing, 2018*), to further examine similarity across layers. Data Mechanics is a data-driven method that can detect clustering in multiple dimensions (i.e., clustering on rows and columns of a multivariate data set) without relying on assumptions of linearity, making it ideal for use with multivariate data derived from complex social systems. The ordering of the rows and columns are shuffled alternately for several iterations to reveal blocks of animals with similar values across the five layer rankings and sets of layers with similar values across the different animals. Shuffling occurs using principles from thermodynamics to assign "energy" to the organization of the network matrix. Network matrices with the lowest energy state are selected as the best ordering for these blocks of subjects (*Guan & Fushing, 2018*). Data Mechanics was applied to the data set of N animals ×5 layer rankings for each group, and the hierarchical clustering tree on the columns was examined to determine which two layers showed the greatest similarity (i.e., clustered together). Data Mechanics was run in R using code obtained from Dr. Hsieh Fushing.

Next, we fitted generalized linear mixed models (GLMMs) to the consensus ranking, with group as a random effect. The consensus rankings ranged from 1 to between 25 and 30, across the seven study groups, with the lowest values corresponding to the individuals with greatest multiplex centrality. These rankings produced a right-skewed data distribution with few animals having Borda ranks near 1 and more animals having Borda ranks >20. We reversed these rankings to range from 0 (lowest centrality) to between 24 and 29 by subtracting each Borda rank from the maximum value in the group which produced an outcome distribution that could be modeled using a Poisson or Negative Binomial family model. Tiers were present in the consensus rankings, and the more populated tiers occurred at the bottom of the consensus ranking (i.e., many individuals had low multiplex centrality). Therefore, the distribution of reversed consensus ranks could be suitably modeled using a Negative Binomial family regression model. Fixed effects in these models included animal age (in years), sex (male or female), dominance rank (the percentage of group members outranked by each subject), dominance certainty (observed range: 0.69–0.98), matriline size (both numerical and categorical versions were tried), rearing history (mother-reared vs. nursery-reared) as well as some interactions between these terms based upon our predictions (e.g., rank × sex). Dominance certainty is a network-based measure of one's 'fit' within the hierarchy and is calculated as the average level of dominance relationship certainty (based on directional consistency of pathways in the aggression network) for each animal in the group. It ranges from 0.5 (ambiguous) to 1.0 (certain) and shows a roughly U-shaped relationship to dominance rank (*Vandeleest et al., 2016*). To facilitate interpretation

of model coefficients, dominance certainty was transformed (subtracted from 1.0) so that a value of 0 reflects complete certainty and 0.5 reflects complete ambiguity. With respect to calculating matriline size, individuals descended from the same female common ancestor at the time of group formation were considered part of the same matriline. Once all models were fitted, we used an Information Theoretic approach to select a candidate set of models to interpret (*Burnham & Anderson, 2002*; *Burnham, Anderson & Huyvaert, 2011*). We calculated AICc scores (Akaike Information Criteria, corrected for small sample size) for each model. We included in the candidate set all models within 4 AICc points of the best model, as this threshold has a high probability of including the best K-L model (i.e., the model that minimizes Kullback–Leibler information loss and thus best approximates reality (*Burnham, Anderson & Huyvaert, 2011*) while also yielding a reasonable number of models to interpret. Model weights were calculated for all models in the candidate set to facilitate understanding of which models have the most support. All GLMMs were run in R using the *glmmADMB* package (*Fournier et al., 2012*).

Additional GLMMs were fitted to the individual network layers that showed the highest correlation to the consensus ranking (i.e., aggression, status signaling, and grooming layers) to determine whether patterns in the relationship between individual attributes and network centrality are different in the multiplex compared to its component layers. We applied the same Information Theoretic approach, described above, for these analyses.

Social network data are inherently non-independent, meaning network measures violate assumptions of data independence, and $p$-values derived from GLMMs cannot be trusted (*Croft et al., 2011*; *Farine, 2017*). To address this, we performed node-based network randomizations on all network layers for all study groups. Node-based randomizations shuffle the attributes of the nodes in the network (e.g., sex, age, dominance rank), which is the relationship we wish to test, and preserves the biologically relevant network structure of the species (*Farine & Whitehead, 2015*). After each randomization, we recalculated the network centrality rankings and fitted the same GLMMs to the randomized versions of centrality as were fit to the observed data. Specifically, 1,000 randomizations were performed separately for each top model being evaluated (1-5 models per outcome variable; see Results), which generated a distribution of model coefficients (one distribution for each term in the model). We used these distributions to calculate significance for each model coefficient for the observed data. Using two-tailed values, the observed coefficient is significantly different from random at $p = 0.05$ if fewer than 2.5% of the random values are larger than the observed coefficient or if greater than 97.5% of the random values are larger than the observed coefficient. All randomizations were performed in R using code developed by co-author MP.

Finally, to determine whether the results from the analysis of the consensus rankings were robust to the threshold used to create rank tiers within each layer, we repeated our analyses with the consensus rankings from a higher threshold (99.5% as opposed to the original 99.0%). Briefly, the analyses of this second set of consensus rankings produced the same best-fit models as the original analysis but with somewhat greater uncertainty because the strength of evidence for the best model was 0.68 as compared to 0.80 in the original analyses. These results are presented in File S1.

## RESULTS

### Pairwise layer correlations

Correlations across pairs of the five layers were relatively consistent across study groups. The strongest positive correlations were found between the aggression and status signaling layers (range: 0.74–0.88; Fig. 2), which is unsurprising given that both aggressive and status signaling interactions are guided by dominance relationships. The Policing layer showed a more variable positive correlation with the Aggression (range: 0.34–0.63) and Status signaling (range: 0.13–0.53) layers, which can arise when some high-ranking individuals do not police conflicts (*Beisner & McCowan, 2013*; *Beisner et al., 2016*). The Huddling layer showed little to no correlation with any layer except Grooming (range: 0.19–0.45; Fig. 2). The Grooming layer showed a variable correlation with the Aggression and Policing layers–moderate in some groups and quite low in other groups. In sum, although some pairs of layers showed a strong correlation, each layer contributed unique information regarding an individual's position.

### Clustering of layers and potential layer redundancy

Hierarchical clustering trees for each study group agreed with the pairwise layer correlations. The aggression and status signaling layers were the two most similar layers, and they were consistently placed in the same column cluster (e.g., Fig. 3), while the other network layers showed more distant relationships. Although the similarity between the aggression and status signaling layers might suggest redundancy, prior multilayer analyses conducted on aggression and status signaling networks from CNPRC social groups demonstrate a critical interdependence between these networks that would not arise if they were redundant (*Beisner et al., 2015*). We therefore kept all five behavioral network layers in the calculation of consensus rankings.

### Attributes and socio-demographic factors associated with consensus rank

GLMM analyses of the consensus ranking revealed two models in the candidate set, with the best model having 79.6% of the model weight (best-fit model: AICc = 3635.3, compared to second-best model dAICc = 2.7). As the strength of evidence for Model 1 was nearly 80%, we report in detail only the results of Model 1 but provide outputs for both models in Table 3. In general, dominant individuals had higher consensus ranks than subordinates (Figs. 4 & 5A). Interactions among sex, dominance rank, and dominance certainty showed that among males (but not females) the effect of dominance rank on consensus rank depended upon the relative certainty (or ambiguity) of the male's dominance rank (Figs. 4 & 5H; Table 3). Having greater dominance certainty reduced multiplex centrality in low-ranked males whereas having low dominance certainty (i.e., greater ambiguity surrounding your fit in the hierarchy) increased multiplex centrality in low-ranked males. In contrast, among high-ranked males, high dominance certainty was associated with higher consensus rank (Fig. 5G; Table 3). Matriline size category also influenced centrality, and its effect differed for males versus females. The consensus ranks of females from medium-sized families (6-10 members) were 1.2 times lower than females from small families (1-5 members) and

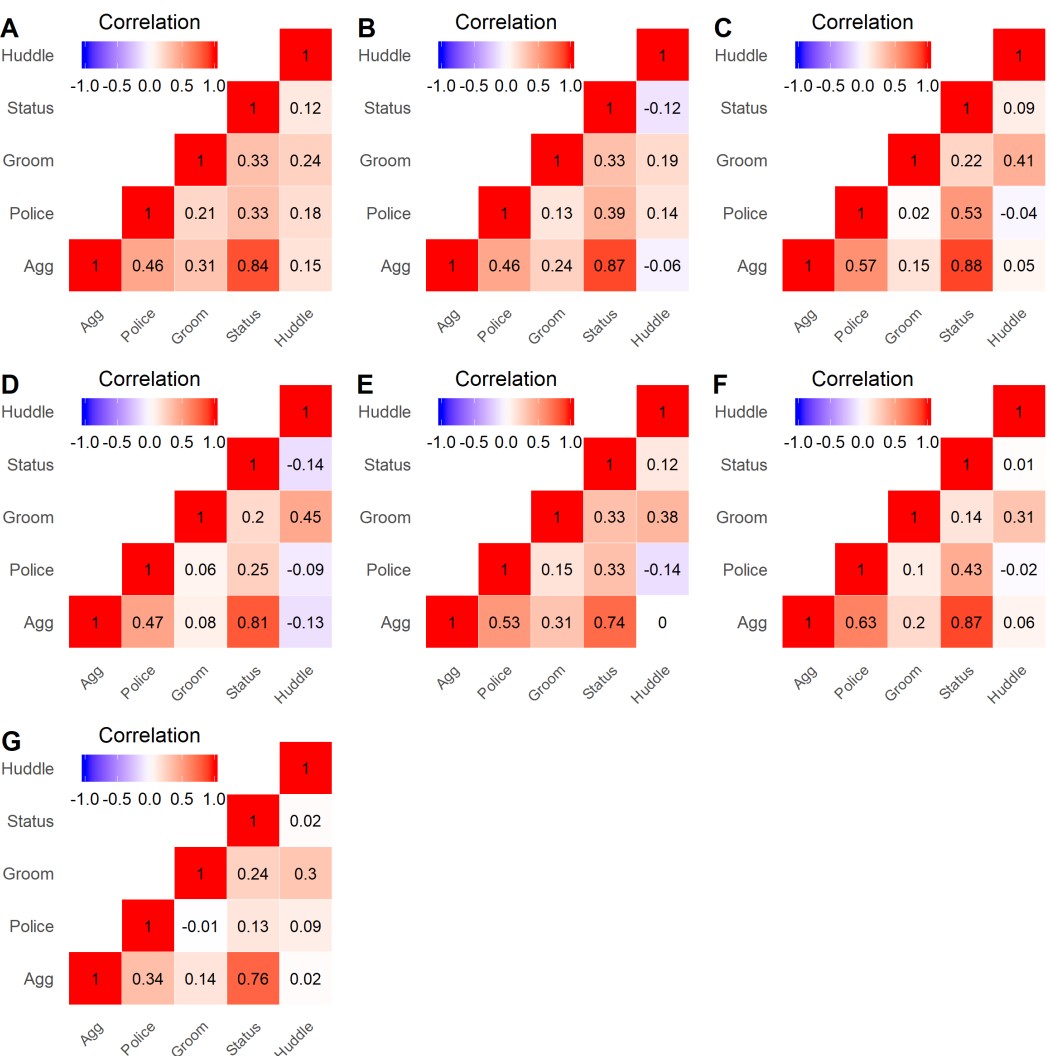

**Figure 2  Heatmap of layer correlations for all groups.** Spearman correlations were calculated for all pairs of network layers, using the centrality rankings of each layer, across all study groups. The lettered labels (A–G) correspond to group IDs.

large families (11+ members, $p < 0.001$); for example, for females of average dominance certainty and rank, the model-predicted consensus rank is 20.0 for females from small and large families compared to 16.7 for females from medium families (Figs. 5E–5F). Matriline size did not affect multiplex centrality among males. Finally, mother-reared subjects had higher consensus ranks than nursery-reared subjects (Fig. 5D; Table 3).

## Aggression network layer

Analyses of aggression layer centrality rank (as measured by eigenvector centrality) yielded two models in the candidate set, with the Model 1 having 87.9% of the model weight and Model 2 having 12.1% (Model 1: AICc =1890.2, compared to second-best model dAIC = 4.0). Model 1 included a 3-way interaction for dominance rank × sex × dominance

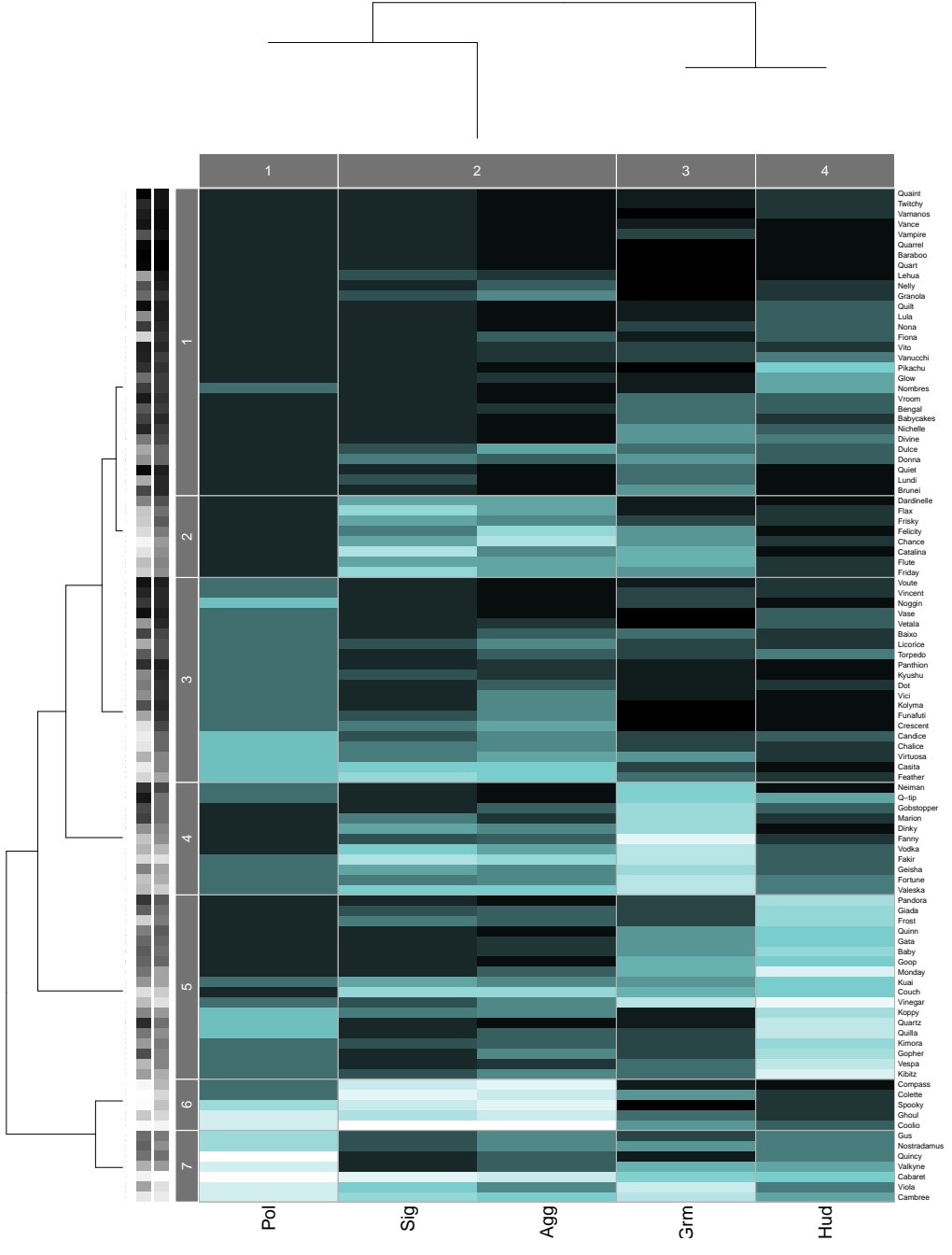

**Figure 3** **Data Mechanics heatmap.** Heatmap showing the block structure revealed by Data Mechanics and the associated hierarchical clustering trees on the columns (layers) and the rows (individuals) for group B. Row clusters group together individuals with similar centrality ranks across the five layers, and column clusters group together layers with similar values across all individuals. Darker colors in the heatmap represent larger values (i.e., low centrality ranks within each layer) and lighter colors represent smaller values (i.e., high centrality ranks within each layer). The grayscale bars on the y-axis indicate animals' dominance rank (left) and Borda rank (right). Lighter colors represent smaller values (i.e., high rank) and darker represents larger values.

**Table 3  Parameter coefficients from the top two models of consensus rank compared against the distribution of model coefficients (Mean ± SD) generated from 1,000 network randomizations.** Coefficients with $p \leq 0.10$ (based upon randomizations) are in bold and italics.

| | Model 1 GLMM coeff's | Model 1 Randomization coefficients Mean ± SD (*p*-value) | Model 2 GLMM coeff's | Model 2 Randomization coefficients Mean ± SD (*p*-value) |
|---|---|---|---|---|
| Intercept | *1.82* | *2.44 ± 0.20 (0.006)* | *1.80* | *2.45 ± 0.21 (0.008)* |
| Sex [male] | *−1.27* | *0.0002 ± 0.26 (<0.001)* | *−1.25* | *−0.01 ± 0.26 (<0.001)* |
| Rank | *1.39* | *−0.02 ± 0.25 (<0.001)* | *1.43* | *−0.03 ± 0.26 (<0.001)* |
| DC[a] | 0.10 | −0.16 ± 1.05 (0.79) | 0.18 | −0.19 ± 1.08 (0.72) |
| Rearing [nursery] | *−0.14* | *−0.001 ± 0.08 (0.07)* | *−0.16* | *−0.0006 ± 0.07 (0.04)* |
| MSC[b] [6-10] vs. [1-5] | *−0.18* | *0.003 ± 0.06 (0.004)* | *−0.13* | *0.002 ± 0.05 (0.02)* |
| MSC [11+] vs. [1-5] | 0.006 | −0.005 ± 0.07 (0.84) | −0.003 | −0.002 ± 0.06 (0.97) |
| Rank × Sex | *1.36* | *−0.007 ± 0.36 (<0.001)* | *1.34* | *0.009 ± 0.37 (0.002)* |
| Sex × DC | *5.04* | *0.03 ± 1.45 (0.004)* | *5.08* | *0.09 ± 1.47 (<0.001)* |
| Rank × DC | −1.03 | 0.16 ± 1.47 (0.39) | −1.23 | −0.20 ± 1.52 (0.33) |
| MSC [6-10] × Sex | *0.21* | *−0.001 ± 0.12 (0.07)* | — | — |
| MSC [11+] × Sex] | −0.05 | 0.005 ± 0.10 (0.64) | — | — |
| Rank × DC × Sex | *−4.89* | *−0.07 ± 2.30 (0.04)* | *−4.84* | *−0.13 ± 2.34 (0.05)* |

**Notes.**
[a] DC = dominance certainty, ranging from 0 (certain) to 0.5 (ambiguous).
[b] MSC = matriline size category (small = 1–5; medium = 6-10; large = 11+ members).

certainty whereas Model 2 included a 3-way interaction for dominance rank × age × dominance certainty and a main effect for sex. As the strength of evidence for Model 1 was over 80%, we report in detail only the results of Model 1 but provide outputs for both models in Table 4. Histograms showing the distributions of model coefficients from the randomizations for Model 1 are available as (Fig. S1). As expected, dominant individuals were more central (had high higher consensus ranks) than subordinate individuals. The effect of rank was strongest for males with greater dominance certainty (i.e., low dominance ambiguity; Fig. S2). Compared to the analyses of the consensus ranks, the factors influencing multiplex centrality were similar to those influencing aggression layer centrality except that matriline size and rearing history were associated with the consensus ranking but not aggression layer centrality.

## Status signaling network layer

Status layer centrality rank (as measured by outdegree) was explained by a single best model (AICc = 1670.6, compared to second-best model dAIC = 15.6) which included dominance rank, sex, age, dominance certainty, and the 3-way interaction rank × sex × age. The model output is provided in Table 5, and histograms showing the distributions of model coefficients from the randomizations are available as Fig. S3). Dominant individuals were higher ranked in the status signaling network than subordinate individuals similar to the analyses of consensus rank and aggression layer rank. The 3-way interaction showed that the effect of age on dominance rank differed for males versus females. Among females, there was no age difference in the impact of dominance rank on status signaling centrality. Among low-ranking males, older males were higher ranked in the status signaling network,

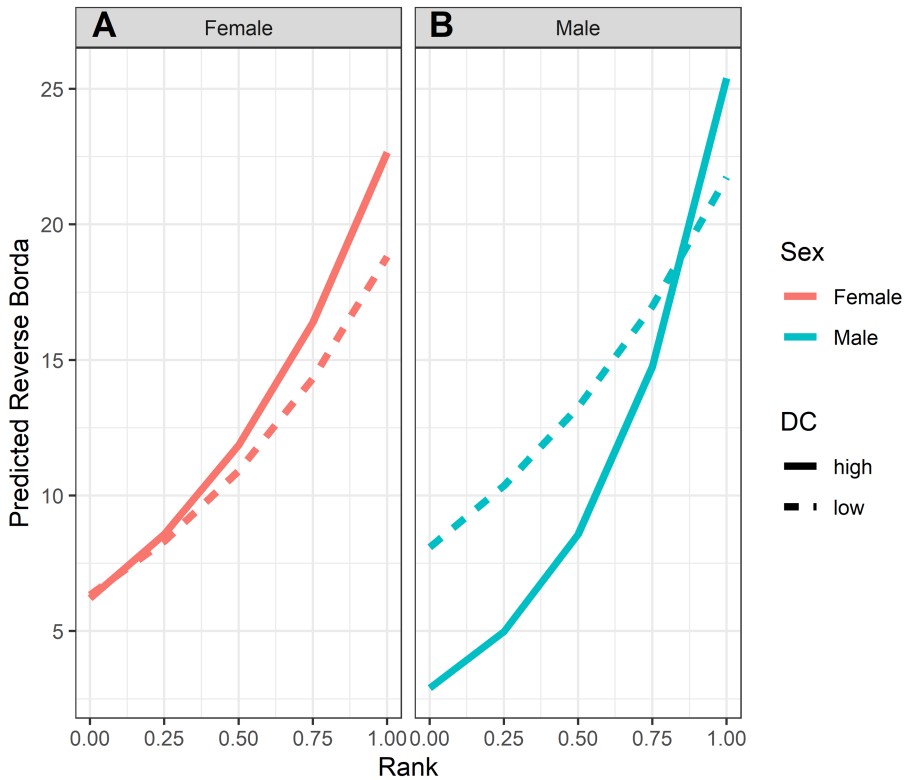

**Figure 4** **Model predicted values of reverse Borda ranks relative to rank, sex and dominance certainty, based upon Model 1.** Predicted reverse Borda ranks for (A) females and (B) males, calculated at two levels of dominance certainty from the observed range of values: high dominance certainty (transformed DC = 0.1) and low dominance certainty (transformed DC = 0.3). Predicted values are based upon Model 1 of reversed Borda ranks.

but among the highest-ranking males, young males were higher ranked in status signaling centrality than older males (Fig. S4). Finally, individuals with greater dominance certainty (i.e., less ambiguity) were higher ranked in the status signaling network.

## Grooming network layer

Analyses of grooming layer centrality rank (as measured by betweenness centrality) yielded five models in the candidate set. Model 1 had 40.4% of the model weight, model 2 had 30.8%, and the remaining three models each had 9–10%. All five model outputs are reported in Table 6, and histograms of the distributions of model coefficients from the randomizations for Model 1 are available as Fig. S5. According to Models 1 and 2, dominant individuals from small matrilines were higher ranked (i.e., more central) in grooming than dominants from larger families, whereas subordinate individuals from large matrilines were more central than subordinates from smaller families (Fig. 6). In other words, the effect of rank on grooming centrality was strongest for individuals from small families and weak for individuals from large families. In all models, females were more central in the grooming network than males. Finally, rearing history was present in Models 1 and 3 but was non-significant based upon the randomizations (Model 1: $p = 0.19$; Model 3:

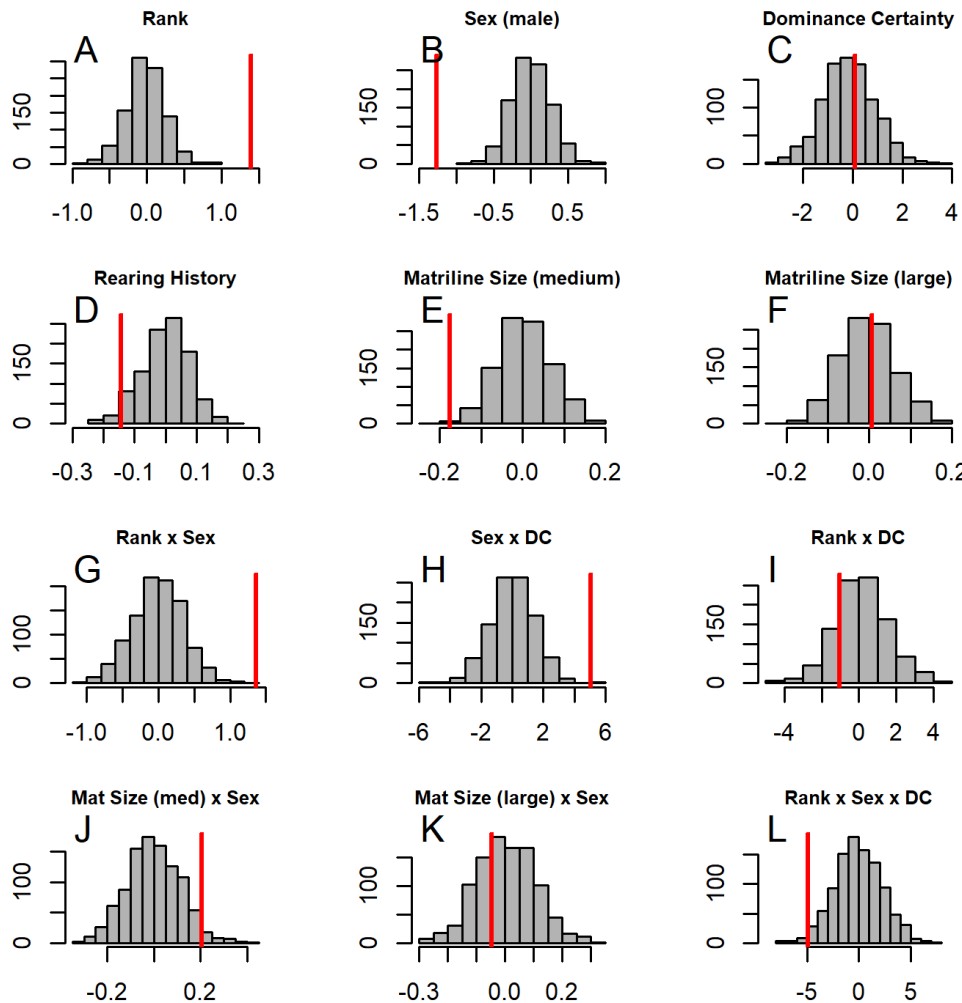

**Figure 5** **Histograms of model coefficients from network randomizations.** Histograms of coefficients for each predictor generated from fitting Model 1 to the network randomizations. For each randomization ($n = 1,000$), all the node labels in the original network (e.g., sex, age, rank) were shuffled; then the same model (reversed Borda rank $\sim$ rank*sex*DC + matriline size category*sex + rearing) was run for each randomized network. The vertical red line shows the value of the model coefficient for each predictor from the observed network data. (A–G) correspond to histograms of the coefficients for the following model predictors of reversed Borda rank: (A) dominance rank, (B) sex (males), (C) dominance certainty, (D) rearing history (nursery-reared), (E) matriline size category (medium-sized families vs. small families), (F) matriline size category (large families vs. small families), (G) dominance rank by sex (male) interaction, (H) sex (male) by dominance certainty interaction, (I) dominance certainty by rank interaction, (J) matriline size category (medium-sized families) by sex (male) interaction, (K) matriline size category (large families) by sex (male) interaction, and (L) dominance certainty by rank by sex (male) interaction.

$p = 0.22$). Thus, in analyses of both the consensus rank and the grooming layer, matriline size impacted centrality, but the precise relationship between matriline size and centrality differed based upon different individual-level attributes. Furthermore, although rearing history was present in the best-fit models for both consensus rank and grooming layer rank, it was only statistically significant for consensus rank.

**Table 4  Parameter coefficients from the top two models of aggression layer rank compared against the distribution of model coefficients (Mean ± SD) generated from 1,000 network randomizations.** Coefficients with $p \leq 0.10$ (based upon randomizations; shown in parentheses) are in bold and italics.

| | Model 1 GLMM coeff's | Model 1 Randomization coefficients Mean ± SD (p-value) | Model 2 GLMM coeff's | Model 2 Randomization coefficients Mean ± SD (p-value) |
|---|---|---|---|---|
| Intercept | *−1.47* | *1.15 ± 0.44 (<0.001)* | −4.04 | *1.15 ± 0.94 (<0.001)* |
| Sex [male] | *−2.29* | *−0.007 ± 0.63(<0.001)* | 0.09 | −0.0001 ± 0.07 (0.22) |
| Rank | *3.81* | *−0.012 ± 0.55 (<0.001)* | 6.72 | *−0.02 ± 1.16 (<0.001)* |
| Age | — | — | 0.26 | −0.001 ± 0.22 (0.25) |
| DC[a] | *5.39* | *−0.04 ± 2.08 (0.01)* | 16.67 | *−0.0002 ± 4.20 (<0.001)* |
| Rank × DC | *−8.04* | *0.12 ± 2.75 (0.005)* | −19.39 | *0.07 ± 5.55 (0.002)* |
| Sex × DC | *8.87* | *0.05 ± 3.02 (0.004)* | — | — |
| Rank × Sex | *2.49* | *−0.01 ± 0.79 (0.002)* | — | — |
| Age × DC | — | — | −1.21 | −0.002 ± 0.93 (0.20) |
| Rank × Age | — | — | −0.29 | 0.001 ± 0.24 (0.24) |
| Rank × DC × Sex | *−8.11* | *−0.01 ± 4.17 (0.04)* | | |
| Rank × DC × Age | | | 1.19 | 0.0008 ± 1.03 (0.25) |

**Notes.**
[a]DC = dominance certainty, ranging from 0 (certain) to 0.5 (ambiguous).

**Table 5  Parameter coefficients from the top model of status signaling layer rank compared against the distribution of model coefficients (Mean ± SD) generated from 1,000 network randomizations.** Coefficients with $p \leq 0.10$ based upon randomizations (shown in parentheses) are in bold and italics.

| | Model 1 GLMM Coeff's | Model 1 Randomization Coefficients Mean ± SD (p-value) |
|---|---|---|
| Intercept | *−0.86* | *0.69 ± 0.32 (<0.001)* |
| Sex [male] | *−2.67* | *0.05 ± 0.50 (<0.001)* |
| Rank | *3.48* | *−0.001 ± 0.42 (<0.001)* |
| Age | 0.03 | 0.002 ± 0.03 (0.31) |
| DC[a] | *−2.93* | *0.05 ± 1.04 (0.004)* |
| Rank × Sex | *2.56* | *−0.07 ± 0.76 (<0.001)* |
| Rank × Age | −0.07 | −0.002 ± 0.05 (0.17) |
| Sex × Age | *0.38* | *−0.02 ± 0.11 (<0.001)* |
| Rank × Sex × Age | *−0.36* | *0.02 ± 0.13 (0.004)* |

**Notes.**
[a]DC = dominance certainty, ranging from 0 (certain) to 0.5 (ambiguous).

## Who occupies top consensus ranks?

Our regression analyses show a strong correlation between dominance rank and multilayer consensus rank (Fig. 7). However, variation around this pattern revealed some individuals that were more or less central than expected. For instance, the precise rank of the alpha and beta males varied across groups. In six of the seven study groups the alpha male and/or beta male appear in the top six consensus rank positions (Table 7). In the seventh group, however, the alpha and beta males' Borda ranks were 15 and 9, respectively. Across the seven groups, there was a lot of variability in which animal was the most central individual: the alpha male (group D), the beta male (groups E & G), an adult male other than the

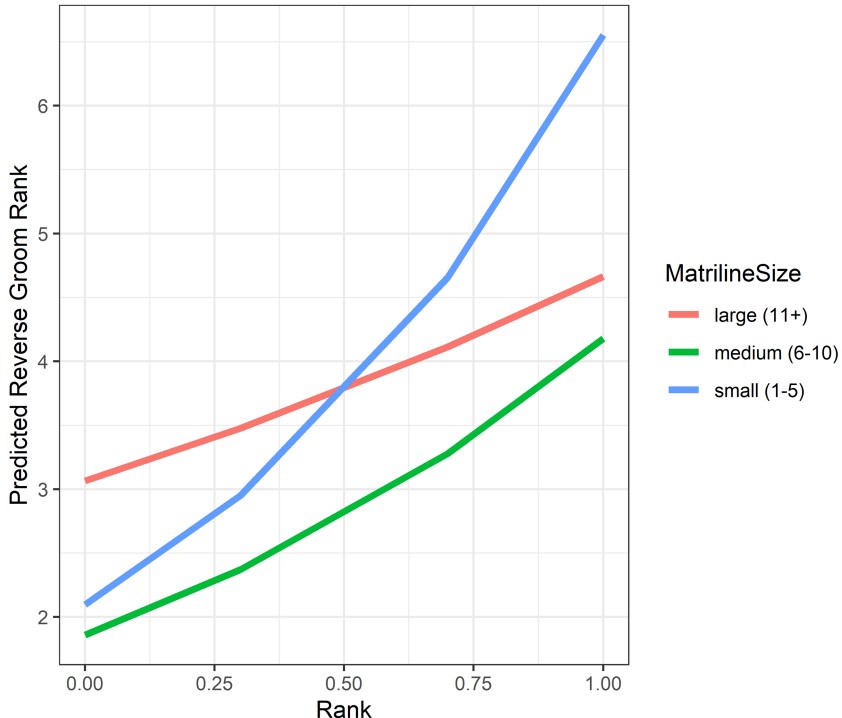

**Figure 6** **Model predicted values of reverse grooming layer ranks relative to dominance rank and matriline size.** Predicted reversed rank in the grooming layer with respect to dominance rank (proportion of others outranked) and matriline size category. Predicted values are based upon Model 1 of grooming layer rank.

alpha or beta (groups A & F), and an adult female from the alpha matriline other than the alpha or beta female (groups B & C). Thus, in four groups (A, B, C, and F), the most central individual was not an alpha or a beta, but another high-ranked animal.

## DISCUSSION

Our analyses demonstrate that the consensus ranking approach (*Pósfai et al., 2019*) for quantifying centrality in a multiplex network yields valuable information about individuals' social connectivity that is both consistent with known patterns for the study species and uniquely different from centrality information from any single layer. The factors influencing rhesus macaques' multiplex centrality aligned well with predictions based upon the literature. High-ranking males with high certainty of their dominance were more central than other males, females from the largest families were more central than females from medium-sized families, and mother-reared individuals were more central than nursery-reared individuals. Importantly, an individual's multiplex centrality, as calculated by our consensus ranking method, could not be reduced to the information from a single layer, as no single behavioral network layer showed perfect concordance to the consensus ranking. Our results confirm the validity of the consensus ranking method for the purposes

Beisner et al. (2020), *PeerJ*, DOI 10.7717/peerj.8712

**Table 6  Parameter coefficients from the top five models of grooming layer centrality rank compared against the distribution of model coefficients (Mean ± SD) generated from 1,000 network randomizations.** Coefficients with $p \leq 0.10$ based upon randomizations (in parentheses) are in bold and italics.

| | Model 1 GLMM coeff's | Model 1 Randomization coefficients mean ± SD (*p*-value) | Model 2 GLMM coeff's | Model 2 Randomization coefficients mean ± SD (*p*-value) | Model 3 GLMM coeff's | Model 3 Randomization coefficients mean ± SD (*p*-value) | Model 4 GLMM coeff's | Model 4 Randomization coefficients mean ± SD (*p*-value) | Model 5 GLMM coeff's | Model 5 Randomization coefficients mean ± SD (*p*-value) |
|---|---|---|---|---|---|---|---|---|---|---|
| Intercept | *0.74* | *1.11 ± 0.12 (0.002)* | *0.72* | *1.12 ± 0.12 (0.002)* | *0.94* | *1.13 ± 0.09 (0.03)* | 0.97 | 1.13 ± 0.09 (0.09) | 0.92 | *1.12 ± 0.09 (0.02)* |
| Rank | *1.14* | *0.04 ± 0.25 (<0.001)* | *1.11* | *0.008 ± 0.25 (<0.001)* | *0.65* | *−0.01 ± 0.11 (<0.001)* | 0.51 | *−0.02 ± 0.14 (<0.001)* | 0.64 | *−0.009 ± 0.11 (<0.001)* |
| Sex [male] | *−0.35* | *−0.005 ± 0.07 (<0.001)* | *−0.34* | *−0.004 ± 0.07 (<0.001)* | −0.32 | *−0.002 ± 0.07 (0.001)* | −0.47 | −0.02 ± 0.12 (<0.001) | −0.30 | *0.001 ± 0.07 (<0.001)* |
| [a]MSC [6-10] vs. [1-5] | −0.12 | −0.04 ± 0.19 (0.41) | −0.12 | 0.03 ± 0.18 (0.45) | *0.25* | *−0.03 ± 0.10 (0.008)* | −0.23 | *−0.03 ± 0.10 (0.008)* | −0.24 | *0.03 ± 0.10 (0.01)* |
| [a]MSC [11+] vs. [1-5] | *0.38* | *0.08 ± 0.14 (0.03)* | *0.39* | *0.06 ± 0.14 (0.03)* | 0.03 | 0.05 ± 0.08 (0.85) | 0.08 | 0.05 ± 0.08 (0.64) | 0.06 | 0.05 ± 0.08 (0.89) |
| Rearing [nursery] | −0.21 | −0.03 ± 0.13 (0.19) | — | — | −0.19 | −0.03 ± 0.013 (0.22) | — | — | — | — |
| Rank × MSC [6-10] | −0.33 | −0.04 ± 0.37 (0.47) | −0.30 | −0.005 ± 0.35 (0.39) | — | — | — | — | — | — |
| Rank × MSC [11+] | *−0.72* | *−0.07 ± 0.30 (0.02)* | *−0.69* | *−0.03 ± 0.29 (0.03)* | — | — | — | — | — | — |
| Rank × Sex [male] | — | — | — | — | — | — | 0.34 | 0.04 ± 0.23 (0.19) | — | — |

**Notes.**
[a]MSC = matriline size category.

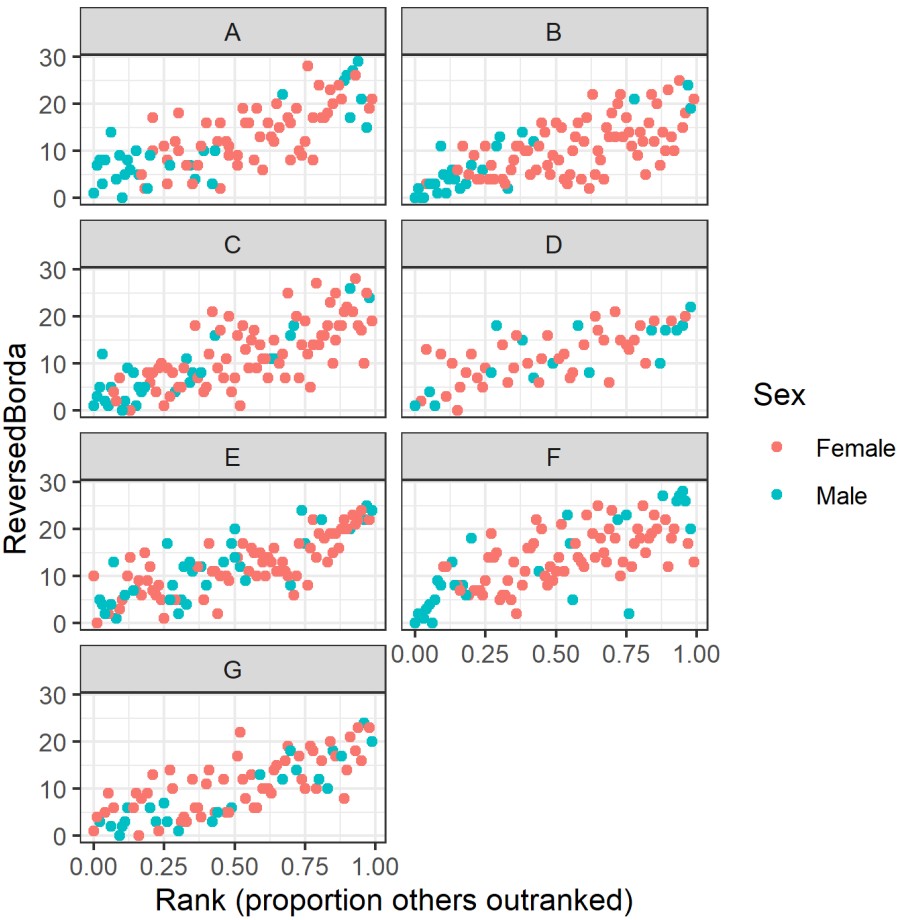

**Figure 7** **Raw data plots of reverse Borda ranks against dominance rank for all study groups.** Reversed Borda ranks (i.e., 1 = low) plotted against dominance rank (proportion of others outranked) for all seven study groups. (A–G) correspond to group IDs. Red dots represent females; turquoise dots represent males.

of assessing multiplex centrality and reveal novel information. We discuss these findings in detail below.

## Analysis of the consensus ranking against socio-demographic factors confirms known patterns

In despotic societies (*Vehrencamp, 1983*), the dominance hierarchy is a central feature of social structure, and one's dominance rank is likely to influence overall social centrality. Consistent with this expectation, we found that high ranking individuals, especially high-ranking males with high dominance certainty, had higher consensus ranks. In other words, they were more central in the multiplex social network. The interactions among dominance rank, dominance certainty, and sex are also consistent with what is known about the study species. Among rhesus macaques, dominant males are considered keystone individuals because they police group conflict (*Bernstein & Sharpe, 1966*; *McCowan et al., 2011*; *Beisner & McCowan, 2013*), receive a disproportionate number of subordination signals (*Beisner et al., 2016*), and are often central in grooming networks (*Sade et al.,*

**Table 7  Which animals occupy top consensus ranks for each study group.**

| Consensus rank position | A | B | C | D | E | F | G |
|---|---|---|---|---|---|---|---|
| 1 | AM[a] | AF | AF | α-AM[d] | β-AM | AM | β-AM |
| 2 | AF[b] | β-AM[e] | AF | AF | α-AM | AM | α-AF |
| 3 | AM | AF | β-AM | α-AF[f] | β-AF[g] | AM | AF |
| 4 | AF | AF | β-AF | AF | AM | AM | AF |
| 5 | AM | AF | AF | AF | AF | α-AM | AF |
| 6 | SAM[c] | AF | AF | AF | α-AF | AF | α-AM |

Notes.
[a] AM, adult male
[b] AF, adult female
[c] SAM, subadult male
[d] α-AM, alpha male
[e] β-AM, beta male
[f] α-AF, alpha female
[g] β-AF, beta female

*1988*; *McCowan et al., 2011*; *Sueur et al., 2011*). However, single layer analyses did not uniformly capture the greater centrality of high-ranking adult males with high dominance certainty. While the best model of aggression layer centrality showed the greater centrality of high-ranking males with high dominance certainty, the status signaling layer analysis showed that *subadults* with high dominance rank were more central than high-ranking *adults*. Subadults' greater centrality in the status signaling layer compared to the multiplex network was likely influenced by of a lack of centrality information derived from the policing and/or grooming networks. For instance, whereas high-ranking *adult* males police conflicts among group members (*Bernstein & Sharpe, 1966*; *McCowan et al., 2011*; *Beisner et al., 2012*; *Beisner et al., 2016*), high-ranking *subadult* males are often natal males from the alpha family (*Koford, 1963*; *Tilford, 1982*; *Beisner et al., 2011b*), and these males are less effective conflict policers (*Beisner et al., 2012*; *Jackson et al., 2012*).

Our results also showed that, among low-ranking males, having low dominance certainty (i.e., greater ambiguity regarding his fit within the dominance hierarchy) improved multiplex centrality, and this makes sense because such males likely stood to gain rank, making their calculated rank an inaccurate reflection of their current status and thus less useful for estimating multiplex centrality. Indeed, the relative certainty or stability of a male's dominance rank can affect the relationship between social status and health in some nonhuman primates (*Sapolsky, 1992*; *Vandeleest et al., 2016*) so its importance for understanding the relationship between social status and multiplex centrality is not unexpected. Finally, we note that although dominance rank was a consistent predictor of centrality in each of the individual network layers, the greater centrality of high-ranking males with high dominance certainty is not trivial because sex and dominance certainty had inconsistent effects across single network layers (e.g., males being central in aggression and status, females being central in grooming and huddling). These results highlight some of the additional insights to be gained by calculating an individual's social centrality across all interaction layers to fully understand their social role and influence.

Natal philopatry (remaining in the home range and/or social group in which you were born) gives kin the opportunity to cooperate or preferentially interact (*Wrangham,*

*1980*; *Stacey & Koenig, 1990*; *Emlen, 1991*). Female rhesus macaques are philopatric and show a clear preference for interacting with close relatives (*Gouzoules & Gouzoules, 1987*; *Bernstein, Judge & Ruehlmann, 1993*). In fact, female kin bias contributes to key features of macaque social structure, such as females' inheritance of dominance rank (due to agonistic support from mothers, grandmothers, and sisters (*Sade, 1969*)). Consistent with the importance of kinship, we found that females from large families (11+ members) had higher consensus ranks than females from medium-sized families (6–10 members). Females from larger families have more kin partners with whom to interact, making it easier for them to be well-connected in grooming or huddling networks. Indeed, the single-layer analyses suggest that the impact of matriline size on centrality is likely through its impact on affiliative interactions because matriline size was a predictor of grooming layer centrality, but not aggression or status layer centrality. However, it was surprising that females from the smallest families (1–5 members) were just as central as females from the largest families. We suspect this nonlinear relationship between family size and multiplex centrality occurs because forming social bonds with non-kin may be a second strategy for being well-connected for females from small families. Female macaques can also form critical social bonds with non-kin (*Datta, 1986*; *Chapais, Girard & Primi, 1991*), and perhaps females from small families form more social ties with non-family members because they have insufficient numbers of family to meet their social needs. Thus, a multiplex approach revealed new information regarding a non-linear relationship between centrality and family size.

Finally, we found that rearing history impacted multiplex centrality –nursery-reared (NR) animals had lower consensus ranks than mother-reared (MR) animals. This is significant because few studies have examined the impact of NR on social behavior of adults living in social groups and results have been mixed. Some studies have found that NR adults are lower ranked than their MR peers when housed together in groups (*Bastian et al., 2003*; *Dettmer et al., 2017*), and others report no rearing related difference in adults' social behavior (*Bauer & Baker, 2016*). NR is well known to cause a number of physiological and behavioral deficits, and these effects have been most consistently shown for young NR animals. NR animals up to 3 years of age exhibit more aggression, less reciprocal social interaction, and an inability to use a social partner to buffer their response to a stressor (*Winslow et al., 2003*) likely due to early alterations in neural and endocrine systems (*Clarke, 1993*; *Sánchez et al., 1998*; *Capitanio et al., 2006*). In this study, we were unable to detect evidence of altered social behavior in NR animals in the single layer analyses but did detect such differences in the multiplex network. These findings offer further evidence that NR can impact adult social behavior (*Bastian et al., 2003*; *Dettmer et al., 2017*), and suggest that the social consequences of nursery rearing are complex and may be difficult to detect by examining a single facet of social life.

## Practical applications for multilayer consensus rank

Multilayer consensus rankings provide an effective and straightforward way to quantify an individual's social role or their impact on the other members of their community. In the study population, consensus ranks revealed that not all alpha and beta males were as socially

central as one might expect and identified other individuals that were surprisingly central. Therefore, multilayer consensus ranks might be a useful tool for management of animal populations, such as those in captivity. For instance, identification of unexpectedly central individuals using a multilayer ranking can benefit the management of captive populations in zoos, research institutions, or sanctuaries, because colony managers must frequently make changes to group membership based on the ever-changing demands of the institution such as breeding, veterinary care, and research. Social instability, and concomitant outbreaks of severe aggression and social trauma, are a known hazard for socially housing rhesus macaques, for example, and removals of key individuals can trigger rank instabilities or a social overthrow (*Ehardt & Bernstein, 1986*; *Oates-O'Brien et al., 2010*; *Beisner et al., 2011a*; *Wooddell et al., 2016*). Identifying socially central, and perhaps influential, individuals can guide management decisions to prevent social instability. Consensus ranks may also prove useful for understanding whether population social structure and animals' typical social roles may differ under different environmental conditions such as rapid anthropogenic change or habitat loss because deforestation, urbanization and ecotourism can significantly impact animal behavior (*Clarke, Collins & Zucker, 2002*; *Lusseau & Higham, 2004*; *Kaburu et al., 2019*; *Marty et al., 2019*).

## CONCLUSION

Multilayer network approaches, such as calculations of multiplex centrality, can offer greater insight into an individual's role in society compared to standard analytical approaches. Current state-of-the-art methods for quantifying centrality in a multiplex do not adequately handle the complexity found in social networks, such as layers whose links differ in density, function and directionality. Here we have shown that a new method that is designed to handle these complexities across layers, consensus ranking (*Pósfai et al., 2019*), appropriately identified which types of individuals tend to be central in a socially complex nonhuman primate (e.g., high-ranking males with high certainty of rank; females from large families) and further highlighted some patterns that might have otherwise gone undetected (i.e., females from the smallest families were as central as those from the largest; NR individuals were less central than MR individuals).

## ACKNOWLEDGEMENTS

We thank our dedicated behavioral data collection team including A Barnard, T Boussina, E Cano, J Greco, M Jackson, A Maness, A Nathman, A Vitale, and S Winkler. We thank Kelly Finn for helpful discussions about multilayer networks. The content is solely the responsibility of the authors and does not necessarily represent the official views of the National Institutes of Health.

### Funding

This work was supported by the US National Institutes of Health (No. R24-OD011136 and No. R01HD068335), the US Army Research Office MURI Award (No. W911NF-13-1-0340), and DARPA Award (No. W911NF-17-1-0077). The funders had no role in study design, data collection and analysis, decision to publish, or preparation of the manuscript.

### Grant Disclosures

The following grant information was disclosed by the authors:
US National Institutes of Health: R24-OD011136, R01HD068335.
The US Army Research Office MURI Award: W911NF-13-1-0340.
DARPA Award: W911NF-17-1-0077.

### Competing Interests

The authors declare there are no competing interests.

### Author Contributions

- Brianne Beisner conceived and designed the experiments, analyzed the data, prepared figures and/or tables, authored or reviewed drafts of the paper, and approved the final draft.
- Niklas Braun conceived and designed the experiments, performed the experiments, authored or reviewed drafts of the paper, and approved the final draft.
- Márton Pósfai conceived and designed the experiments, performed the experiments, prepared figures and/or tables, authored or reviewed drafts of the paper, and approved the final draft.
- Jessica Vandeleest, Raissa D'Souza and Brenda McCowan conceived and designed the experiments, authored or reviewed drafts of the paper, and approved the final draft.

### Animal Ethics

The following information was supplied relating to ethical approvals (i.e., approving body and any reference numbers):

The Institutional Animal Care and Use Committee of the University of California Davis approved the research (IACUC protocol #18525) from which the behavioral data were taken.

### Data Availability

Individual-level attributes (e.g., sex, age), socio-demographic factors (e.g., dominance rank & certainty, matriline size), the network layer rank orders as well as the multilayer consensus ranking generated from those layers for all seven study groups are available in a Supplemental File.

## Supplemental Information

Supplemental information for this article can be found online at http://dx.doi.org/10.7717/peerj.8712#supplemental-information.

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
