# Peer review of "A multiplex centrality metric for complex social networks: sex, social status, and family structure predict multiplex centrality in rhesus macaques"

_PeerJ, doi:10.7717/peerj.8712_

## Round 0.1 · original submission · Major Revisions

The reviewers generally commented positively on the manuscript and provided constructive feedback that will help to further strengthen the manuscript. In particular, the authors should reconsider the statistical approach that was used to analyse the networks and improve the quality of the figures.

·

Basic reporting

The authors present an interesting new approach for estimating multilayer centrality that can handle common attributes of animal social networks: namely, network layers with different properties and functions. The authors then validate this approach by applying it to a well-understood social system, that of rhesus macaques. I found the paper to be clearly organized and well-written, with only occasional grammatical errors (noted under General comments). My main concerns are with the statistical approach, which I discuss further below.

The introduction does a nice job of introducing multilayer network approaches, describing the situations in which they may be helpful, and explaining how the currently available measures for describing multilayer centrality are often inadequate for animal social networks. I have only a few points where I feel additional clarification could be helpful:

L46, L63-64, L79-80: Is all social life complex? Are all animal social networks complex? I feel it would be helpful if the authors clearly state what attributes they believe make for a complex social system or social network.

L81-137: Following on from the above point, although the authors clearly present the rationale and potential advantages of using multilayer networks, are there instances in which the added complexity introduced by these approaches is not warranted?

L178-179: While this point becomes clearer later on (e.g. L330-339), a brief discussion or example here of what makes a particular centrality measure more or less appropriate for a given network would be helpful.

Table 2: Please define what network density is, and how it is calculated.

Figure 1: The caption should describe each portion of the figure (a, b, & c) separately.

Tables 3-6: Please provide standard errors and confidence intervals for model coefficients.

Figures 6: Can confidence intervals be included here?

Figure 8: The caption should indicate what the plotted line and grey regions represent.

In addition to the raw data that has been provided, the authors may wish to consider uploading the network data itself if it’s not already available.

Experimental design

L322-325: Was the severity of aggressive interactions (L290-293) included in the network, and if so, how was this done?

L326-328: Are grooming interactions generally asymmetric in rhesus macaques (e.g., A grooms B, but is not groomed by B)? If so, would it make sense to also treat this network layer as directed?

L335-339: Could more information be provided about how centrality was determined for the policing network? For example, assuming in-degree and out-degree are simply added together, one could envision peculiar situations such as individual A (policed 7 others, but was never policed itself) and individual B (never policed others, but was itself policed 7 times) having the same centrality.

L360-364: Although I appreciate the authors’ point regarding the difficulties of deciding the relative importance of different layers, I imagine that some researchers interested in using consensus rankings will have reason to suspect that particular network layers will be especially important. Are there potential modifications to this method that would allow such weightings to be incorporated?

L418-425: Burnham and Anderson (2002) caution against using arbitrary cut-offs, such as ΔAIC < 4 (see also Anderson, 2004, Model based inference in the life sciences; Burnham et al., 2011, Behav. Ecol. Sociobiol. 65: 23-35). This is particularly important for models with ΔAICc in the range of ~2 – 7; these still have some empirical support, and so should not be automatically discarded. If the authors wish to reduce their model set, perhaps the concept of a confidence set can be employed (Burnham & Anderson, 2002, pg. 196; Anderson, 2004, pg. 121).

With regards to removing more complex models with similar AIC scores to the top model, AIC already takes into account parsimony. As such, I think it would be better to leave those more complex models in the model set. That said, Burnham and Anderson (2002), pg. 131, do present a specific case where more complex models with ΔAIC in the range of 0 – 2 may be disregarded, which may be where the authors took their guidance for the pruning described here.

Including the more complex models (as well as models with ΔAICc > 4) may increase model selection uncertainty, but techniques for multi-model inference can be employed if this uncertainty turns out to be substantial.

Validity of the findings

L454-462: While I appreciate the authors’ justification for including both the aggression and status signalling layers when calculating consensus ranks, I think it is still important to test whether the main results change appreciably if one or the other of these two layers is excluded. If both layers are actually providing similar information about an individual’s social position, then including both in the consensus ranking risks inflating the importance of this information (especially since each layer contributes equally to those rankings).

L463-515: P-values obtained from GLMMs using network data generally cannot be trusted. These statistical models assume independent observations, but individuals’ network positions are not independent of one another (see Croft et al., 2011, TREE, 502-507; Farine, 2017, Methods Ecol. Evol., 1309-1320). To address this issues, a null network model can be constructed and incorporated into a GLMM (Farine & Whitehead, 2015, J. Anim. Ecol., 1144-1163). Please address this issue.

L473-474: Statistical information regarding the contrast between females from large families versus those from medium families is not presented here or in Table 3.

L503-515: The analysis of the grooming network provides a nice example where multi-model inference would be helpful. The model probabilities indicate substantial model selection uncertainty, with all five models receiving some empirical support.

L533-536: Earlier (L230-236), the authors reviewed literature indicating why females from large families were expected to be central, but was it predicted a priori that females from small families would also be central? Given that in L583-584 this finding is noted as surprising, perhaps only the former result should be included here in the list of expected results.

L536-539: I believe it would be good to acknowledge here that certain network layers (aggression, status signalling) provided a better match to the consensus rankings than others.

L613-614: Although the discussion immediately following this statement does a nice job of illustrating the potential of multilayer approaches to better characterize social patterns, I suggest tempering this first sentence somewhat. Consensus rankings certainly indicated differences in centrality compared to analysis of the single layer networks. However, whether those individuals with high consensus rankings are actually more important than individuals that are central within only a single network (i.e., in terms of their contribution to social functioning, group stability, transmission dynamics, etc.) remains to be experimentally tested.

Additional comments

L31-32: State the species here.

L40: Presentation of the statistical results here are somewhat confusing—e.g., females from the largest families were found to have higher centrality, yet the comparison between large and small matrilines is not significant and the comparison between larger and medium matrilines is not presented.

L128: “be a popular grooming partner”

L151: “node-level measures”. Also, should “network” follow “multiplex”?

L218-219: remove “are”

L269-270: “and the laws of the United States government.”

L467-510: Would it make more sense to discuss rankings as higher or lower (rather than better or worse)?

Reviewer 2 ·

Basic reporting

In the manuscript “A multiplex centrality metric for complex social networks: sex, social status, and family structure predict multiplex centrality in rhesus macaques (#38664)”,
the authors present an original and interesting method to measure multiplex centrality which allows using different centrality metrics to account for each layer depending on its dynamics and functionality.

Overall, the paper is very well written and organized, making it easy to follow and understand. It is a self-contained manuscript, well referenced and supported by tidy helpfull tables and figures. The introduction is divided in helpful subsections providing relevant background properly supported by accurate references. Enough raw data is provided to replicate the core analyses presented. The results directly respond to the objectives and hypotheses presented.

Experimental design

This is an orignal study is based on a solid data set including information from 5 interaction types between members of seven captive groups of Rhesus macaques, a well-known study species. The objective of the study is clearly defined and highly relevant to current demands for analytical tools to develop social network analyses that better respond to biological questions based on empirical data. The authors introduce a novel method to infer individual centrality based on multiple network layers (interaction types/contexts), and they also include validation of the method taking advantage of the known social organization and dynamics of the study species. The methods are well described and seem rigorous and carefully thought and carried out. The raw data provided presents the centrality ranks for each multiplex layer allowing for replication of the calculation of multiplex centralities and comparison with single-layer results.

Validity of the findings

Results are clearly presented and well supported by tables and high quality figures properly labelled (particular comments on figure captions are shown below). Although it is not the first method to calculate multiplex centrality, it has characteristics that are absent in other known methodologies, thereby expanding the possibilities of analysis, particularly when using different centrality measures across layers may be a better way to address the problem or system of interest. I find it relevant and useful, not only for the field of primatology and for studies of social structure at large, but also as an interesting contribution to social network analysis in general. The results are nicely discussed and the conclusions are well stated and directly derived from the results.

I only have two concerns regarding the methodology which I would like the authors to discuss to some extent. First, is the use of GLMMs to analyze the centrality ranking data derived from interactions not violating independence assumptions (see Farine and Whitehead 2015 "Constructing, conducting and interpreting animal social network analysis")? If so, how could this affect the methodology proposed? Could it be better to consider methods like ERGM devised specifically for netwrk data? My second concern is related to the robustness of the centrality results. From what I understand, the analyses presented did not involve randomizations or any other method to estimate uncertainty about the centrality ranking results for each individual layer and for the multiplex centrality. If this is so, how could the method be complemented to account for effects resulting in small differences between centrality rankings? How do we know that the differences observed are not random?

Additional comments

The following are specific comments/suggested edits directly related to specific lines from the text or figure captions:

L82. I suggest avoiding the term network when referring to the layers of the multiplex network. Instead, use layers as you do further along the text.

L84-86. I suggest define the meaning of network layers (what they represent) before defining multiplex networks.

L118-119. Edit to: “…can further give insights into an individual’s social position/role within a society”

L128. Edit “partner” to “partners”

L129. Edit animal to animals


L218. Edit: remove “are”

L325-328. So grooming direction is not influenced by the dominance ranking? By defining the grooming layer as non-directional, this suggests that the information about directionality is not considered as relevant to centrality ranking as, for instance, aggression and dominance signaling. I this something well established in Rhesus macaques? Or why disregard directionality when the method devised allows for its inclusion and it may convey additional information? Please explain a bit more.

L334-335. Given that eigenvector centrality is less intuitive than other centrality measures, I suggest adding a reference to this statement or a bit more explanation on how it is better for finding positions in hierarchical systems.

L380. Edit to: “… how much of the variance of the untiered centrality values is explained by the tiered centrality values.”


Comments to figures.

Figure 1. Please add a bit more explanation to the caption, particularly referring to the three parts of the figure labeled a-c. Also, please mention what is indicated by bar color in 1b and make some mention of the y axis units in 1c (1-r2)

Figure 8. Please mention the confidence bands shown in the figures including confidence level.

·

Basic reporting

The article is largely clear and well written. In places there are quite technical concepts that need to be better explained to make them accessible to the intended audience of the article (have detailed these in my line-by-line comments in the general comments box).

Use of literature is good. One paper that should probably be added in discussing the use of multilayer networks in animal behaviour is:
Finn et al. 2019. The use of multilayer network analysis in animal behaviour. Animal Behaviour, 149, 7-22.

Article structure is generally clear. I personally would prefer to see some of the initial sub-section of the Methods at the end of the introduction to help set up the hypotheses/predictions of the study at the end of the introduction. The discussion is a little repetitive at times (highlighted in my line-by-line comments below)

I would provide confidence intervals around model predictions rather than indicating significance in the Tables. Using p values and AIC together has some strong opponents and it is often best to use whether the 95% confidence interval crosses zero or not (in the model of interest or an averaged model) as a measure of the statistical significance of the result

The figures need considerable improvement - simply plotting model predictions without either the raw data points or an indication of the error around these predictions is unhelpful.

Experimental design

The research questions are well defined but it would be good to see hypotheses and predictions set out more clearly in the introduction.

Data collection methodology appears rigorous and has suitable ethical approval, and in general the methods are described in a good level of detail (with some minor exceptions detailed in my general line-by-line comments). The one general exception to this is that I would say that the Methods is very light on describing what software was used and how it was used to conduct various aspects of the data analysis.

I worry a little that different network metrics have been calculated for different layers - not so much for the calculation of the consensus rank (where it has been justified) but for the calculation of correlations between the ranks of layers (Fig 2), as betweenness centrality and eigenvector centrality might well be expected to not be correlated anyway creating a distinction between the correlation values calculated that does not depend on differences between the layers. I wonder whether a matrix correlation between different layers would also be of value.

I am concerned by the failure to account for the non-independence of observations in the network in statistical inference from the models fitted. In general animal social networks that fit LMs or GLMs use randomisations to test statistical significance (e.g. https://www.sciencedirect.com/science/article/pii/S0169534711001455) or use MCMC approaches to fit them (e.g. https://www.nature.com/articles/s41598-017-18104-4) to deal with the issue of non-independence, and I feel that using one of these alternative methods is important here unless the authors have a strong justification as to why they think the non-independence issue does not arise in this dataset.

Validity of the findings

Results are fairly well explained given that they are fairly complex. I would suggest being consistent with how you treat variables that are in the top model but with confidence intervals that cross zero (i.e. are statistically non-significant). I think the simplest and clearest way to do this is to state that they are in the top model but also say that they are non-significant universally (avoiding language like tended for example).

I would also suggest using higher rather than better when discussing consensus ranks (this comment applies elsewhere too)

For the results for separate network layers it would be worth reminding the reader which network metric was used as this might affect the results. This should also feature as a discussion point. Eigenvector centrality and betweenness centrality will indicate different social roles.

The discussion is rather long but very well written, I only have minor comments detailed below.

Additional comments

Some line-by-line comments:

Abstract
L22: "welfare" is a slightly odd choice here given you haven't yet specified between wild and captive animals
L28-32: Would suggest breaking up this sentence. Something along the lines of "Here we validate a new method for quantifying multiplex centrality called consensus ranking by applying this method to multiple social groups of a well-studied non-human primate, the rhesus macaque. Consensus ranking can suitably..."
L33: Suggest changing "predictive of" to "correlated with"
L37-42: I am not a huge fan of p values in abstracts. Your certainly need to make sure you are consistent here though if you do use them, some results here have p values and others don't

Introduction
L52: Suggest deleting "However, " to avoid having a however immediately followed by a yet, which reads strangely. Otherwise nice opening paragraph!
L84-88: I think it is best to start here with the technical definition here that a multiplex network can only contain inter-layer edges that connect the same actor in different layers. You can then go on to say that this normally equates to different layers representing different sets of behaviour interactions between the same or a similar set of individuals
L91-93: Here and elsewhere in the introduction you provide a good explanation of why you use multiplex networks versus single networks, but you also need to explain why multiplex networks are more informative than simply aggregating all of the layers to complete this motivation
L117: Here and elsewhere you need to make sure you are clear that "multiplex centrality" consists of multiple possible network metrics - too often it is sounding like a single measure in its own right.
L123: Another example where it is important to distinguish the advantage of multiplex to that of aggregating networks
L151: "node-level measureS of connectivity" and "in a multiplex NETWORK broadly"
L155: In my opinion you will have to provide a more accessible explanation of a tensor for a zoological/ecological audience or avoid using the terminology
L178-179: This needs a little more context/explanation
L184-188: First, I find it a little odd that the wording of this is almost identical to the abstract. I would suggest making a similar swap to that suggested earlier to make it easier to read.

Methods
L210: Would suggest linear rather than clear as you have used clear also in the previous sentence
L306-307: Not immediately clear what this means or whether that is deemed sufficient - a little more detail here would be helpful
L325-328: It seems strange to me that direction is deemed unimportant in grooming networks given the evidence in other study systems of asymmetry in these relationships - is this something particular to macaques? Potentially a more careful explanation would be of value here.
L333-334: It is not clear to me why betweenness makes a good measure of cohesion? Surely a measure such as closeness has a much clearer link to cohesion of groups. Individuals in some network structures can high betweenness when acting as bridges between communities despite having low cohesion to both. As such, I am struggling a little bit with this justification to use betweenness.
L336: Are in-degree and out-degree of the policing layer summed separately or together? It strikes me both have rather different meanings behavioural and summing them together could be problematic and also lose lots of information about the social role of an individual.
L360: On point b), it strikes me that a nice strength to this new measure when calculated this way is that by changing the weighting used when summing the ranks you can actually vary how much influence each layer has, which may be valuable in some contexts. Perhaps worth discussing somewhere?
L364: One potential weakness of this centrality measure in a multilayer (from what I can see) is that the ranking in each layer is done separately and so it doesn't incorporate the multiplexity of the network very well. Therefore an individual connected to the same 10 individuals in layer A and layer B will have a higher ranking than one connected to a different 7 individuals in A and B (and therefore 14 individuals overall). Other measures of multiplex centrality also have this issue but I think it is important to mention or caveat somewhere. If this could be factored into this measure too at some point down the line it would be great.
L365: This sub-section is one that I think I would struggle to replicate with the current level of detail. I think it particular it would be valuable to provide more information on the practicalities of how the clustering algorithm was applied and also how the output of the clustering algorithm is used.
L397-402: This is another approach that needs to be outlined more carefully to make it fully replicable. More help is needed in how the approach was applied and stronger links to the biology of the study system would help understand why it was used.
L411: Please make it more clear and intuitive why this transformation of the data is suitable for use in a negative binomial model.
L413-414: Please provide methodology for how dominance certainty was calculated and also how matrilines were defined to enable matriline size to be calculated. Dominance certainty is particularly important to understand how it fits in with aggression and status signalling - is it a separate measure or does the calculation of dominance certainty use these variables?
L415: What do you mean by all interactions? Up 3-way? Or were up to 6-way interactions fitted? This is currently very unclear.
Would be helpful to define the alpha/significance level somewhere in the analysis subsection of the methods

Results:
L443: Suggest changing "highest" to "strongest positive" to be clearer with direction
L446: Suggest "more variable POSITIVE correlation" to ne clear with direction
L449-451: Suggest adding some numbers/model predictions for this result
L461: Suggest changing to "in the CALCULATION OF consensus"
L467-468: This result is not immediately apparent from the model results (main effect term is NS + there is an interaction anyway) so you need to be clear with where it comes from
L484-485: Avoid this (see comments in Box 3)
L495-501: A supplementary figure or some mode estimates/standard errors to support this verbal description of the results would be helpful

Discussion:
L533-536: I feel more careful wording here to avoid equating general effect with characteristics of "the most connected individuals"
L573: I would suggest editing to "In fact, kinship among females contributeS to key..."
L586: Would suggest adding a qualifier at the end of this sentence e.g. "...for being well-connected for females in small families". Would also suggesting adding a "can" at the start of the next sentence "Female macaques CAN also form"
L613-614: More careful wording is needed here. You have shown you get different results, but succeed implis there is a right/wrong which I would disagree exists. I think you can say that the multiplex provides a more holistic overview without claiming it is the correct approach.
L617-618: This subadult result didn't really come through clearly in the results section - be more clear linking age to subadult/adult in the results too for consistency.
L620-621: Again, this comparison isn't entirely fair - I imagine the multiplex doesn't reveal everything we know either, perhaps again temper this a little to say that the multiplex has provided a more integrated/holistic view of social structure etc.
L623-634: I find a lot of the ideas in this paragraph rather repetitive of things that have come before, especially in this latter section - it loses the flow of the discussion somewhat.
L638-639: I would suggest avoiding this
L662-663: This seems rather repetitive from earlier in the paragraph
L668-671: This seems a rather weak ending to the paragraph and I am unsure what is meant by talent in this context? I I wonder whether this sentence could be deleted.

Tables 3-6: Would suggest providing an indication of confidence intervals around model estimates (perhaps instead of p values as highlighted in Box 1)

Figure 1: Please provide more helpful and detailed figure legend
Figure 2: Any reason why this couldn't be multi-panelled to provide equivalent data on all of the groups?
Figure 3: Please provide more helpful and detailed figure legend
Figure 4-7: I would strongly suggest you add confidence intervals and preferably raw data to these plots. They also require more helpful and detailed figure legends
Figure 8: Would be good to see a version of this showing all 7 groups. The figure legend also needs to be more detailed e.g. what are the lines/polygons plotted here and why have they been plotted? Why is dominance rank on a different scale to the other figures?

I hope my comments are useful in improving the manuscript.
Matthew Silk

---

## Round 0.2 · Minor Revisions

While the revised manuscript has improved considerably, the reviewers have made additional comments that the authors should use to further improve their manuscript.

In their response letter, the authors state that they would rather not release the actual network data itself as publications are on-going for this dataset. The authors should update the data availability statement accordingly, e.g. by indicating that these data and/or custom scripts will be supplied upon reasonable request.

·

Basic reporting

The manuscript is clearly written and structured well. The Introduction provides a useful overview of the potential benefits of adopting a multilayer approach, clearly explains why consensus ranking may be useful for situations other multilayer centrality metrics may be ill-equipped to handle, and shows how rhesus macaques offer a useful testbed. Coverage of the literature is good throughout.

The figures and captions have been much improved since the initial submission; I have only minor suggestions in the general comments.

The raw data needed to obtain the GLMM parameter estimates has been provided.

Experimental design

The manuscript presents a novel metric for evaluating centrality in multiplex networks (consensus ranking). This metric successfully captures known features of rhesus macaque social structure, as well as reveals novel aspects missed by single-layer approaches. As consensus ranking can be used in situations where network layers differ in topology and function, it may prove a valuable addition to network analysts’ toolbox.

Data collection protocols and statistical analysis are sound and appropriate ethical approval was obtained. Network randomizations are now integrated with the GLMMs in order to evaluate statistical significance, correcting a major weakness of the initial manuscript.

The methods are generally explained well. I appreciate the additional information the authors now provide for certain methodological decisions (e.g. treating grooming interactions as undirected, summing in- and out-degree in the policing layer).

Validity of the findings

The adjustments made to the Discussion are welcome. The paper flows very nicely, with conclusions logically stemming from the results.

Additional comments

L38: I suggest avoiding abbreviations in the abstract (even a common one such as this).

L61: Include scientific name at first mention.

L127: Delete “in”

L157: Consider describing what betweenness captures without using network jargon such as ‘shortest path’ (as was done nicely for the other metrics here). Perhaps something along the lines of: “the tendency to link together otherwise weakly connected individuals/groups”?

L330-333: As it is brought up earlier (L293-296), I suggest clarifying here that information on aggression intensity was not included in the aggression layer.

L344: In the authors’ response letter, more justification is provided for the use of betweenness. I think it would be helpful to include this information in the text itself; many readers may primarily think of betweenness as a means to evaluate importance for flow/transmission processes, rather than as a measure of social cohesion.

L346-349: I appreciate the additional explanation provided for how the policing layer is treated. I wonder if L325-327 might even be better placed here. Otherwise I fear readers may miss the fact that the policing layer is trying to capture information on valued social relationships (including victims that need defending) rather than simply describing power differences.

L399: “explained by the tiered”

L429: Define the abbreviation ‘GLMM’ here prior to its use later.

L444-448: Consider breaking this sentence up to improve readability.

L525-527: I suspect the authors are correct that the difference in consensus ranks between females from medium and large families is statistically significant, given there is essentially no difference between females from small and large matrilines. However, would it not be relatively simple to change the baseline level and obtain the corresponding p-value?

Figure 1 caption: Thank you for including more information here; the figure now stands on its own. While it is discussed in the text, I suggest including here what the y-axis in 1c represents (i.e. that r^2 is the correlation between the original and tiered centralities, and that the horizontal line indicates where 99% of the variance is explained).

Figure 5 (and similar figures): It would be helpful to include “Sex (male)” here and elsewhere.

Figure 7: Consider re-drawing this figure to match the visual style of your other figures (e.g. Figure 4).

Table 3: The caption says three models are presented. I also suggest defining MSC beneath the table.

Reviewer 2 ·

Basic reporting

No comment

Experimental design

No comment

Validity of the findings

No comment

Additional comments

After going over the rebuttal and the revised paper, I consider most of my concerns were properly addressed by the authors. Along with the changes made in response to the comments from the other reviewers, I consider the paper was nicely improved and presents very interesting and useful methodological information. The paper is well structured and presented with clear language which makes the study easy to follow. I only have very few observations to the text which I list below. Otherwise, I think the paper is basically ready for publication and I’ll be excited to read the final version.


L64. Edit to: “interact in a variety of contexts…”

L66. Edit to: “…to study…” or “…for the study…”

L127. Edit to: “…heterogeneous…”

Figure captions:

Figure 3: for the dominance and Borda ranks shown in the red-yellow scale, please indicate which color indicates higher and lower ranks.

·

Basic reporting

The revised article is well written and structured. Figures are generally good (although some room for improvement - see my general comments to the authors)

Experimental design

The research question and hypotheses are clear and the methods are largely excellent. Some additional details required in some places to improve clarity further (see my general comments to the authors)

Validity of the findings

I have one new concern related to the analysis that arises from it not being clear how dominance ranks are calculated (apologies for missing this before). If the same data are used to calculate dominance rank (an explanatory variable in several analyses) and the agonistic network/multiplex network and position of individuals within it (the response variable) this could be problematic in terms of model assumptions (although I lack sufficient knowledge to know if it problematic in a practical perspective). I suspect it is potentially a bigger issue in the analysis of the agonistic network IF the same data are used (it may well be that they aren't). This is also outlined in my comments to the authors.

The results seem robust and the discussion/conclusion well-written and clearly stated.

Additional comments

The authors have done an excellent job revising the paper. I have a few remaining concerns and some very minor suggestions highlighted here.

One potential concern that it seems was not picked up on previously (I have tried my best to search for it in the response/original comments and not found it) is how the dominance ranks were calculated. I am assuming they were calculated from a separate dataset than that used to construct the agonistic network used in the study? Currently there is insufficient detail to know this for sure. Otherwise there are potential issues caused by an explanatory variable and response variable not being measured independently from each other (i.e. the dominance ranks that are being used as an explanatory variable are calculated from the same). I know this violates statistical assumptions if it is the case but am not knowledgeable enough to comment on how problematic it is from a practical point of view. It would be a particular issue in the analysis of the aggression layer (although arguably still exists in the full multiplex analysis) and only if the same data was used to construct the networks and calculate ranks. I am hoping/anticipating this is purely a problem of not knowing for sure how dominance ranks were calculated.

Another interesting conundrum is the use of AICc for model simplification prior to the use to randomisations for statistical inference. There might be some situations where this is problematic, but I suspect this isn’t the case for the node-label swaps conducted here (more likely when there is structure/bias in how network data is collected and swaps are to the raw data instead). While I tend to fit randomisations to a full model, I think what the authors have at the moment is a sensible/practical solution to using randomisations alongside a complex model. However, I lack statistical and study system knowledge to be confident that its definitely without caveats. No need to address this comment per se, but potentially worth the authors thinking how removing potential variables using AIC before formal analysis using randomisations might have an impact given their knowledge of the system.

For Figs. 4, 6 and 7 I still think it is important to illustrate something beyond just the model predictions – showing the raw data or group means might be helpful here? While, I think Fig. 4 and 7 are likely fine as they are as they show enough information to be helpful, Fig. 1 as it stands would be better removed and the numbers provided in the main text (as this would be an easier and more accurate comparison). However, Fig. 4 seems like the easiest figure to include some aspect of raw data (group means perhaps?). Apologies for my contradictory suggestions on the previous draft that made this point hard to address.

In L330-338 it is important to say whether weighted or unweighted networks were used and if the former then clarify how weights were calculated. Then when talking about measures be clear whether they are weighted/unweighted measures.

In L464-481 you need to make sure that is clear that randomisations are conducted separately for each top model identified. Which models are used for randomisations is only clear once you have got to the results section currently.

Minor suggestions:
L51-52: This sentence is a little unclear and I feel could be improved by expanding on the point made a little (and perhaps with citations)
L87: This risks confusion as not all layers need necessarily include all of the actors. Deleting “all layers have the same node set” and editing “and this” to “which” would provide an identical definition without this confusion
L116: Suggest “like MANY network measures” as some measures do only account for direct connections (e.g. degree)
L154: Might be too fussy but would suggest “extent” rather the “degree” as it prevents potential confusion with the use of degree as a specific measure
L156: Make sure the definition of degree includes the idea that is the number of or summed weight of all connections of an individual
L157: “eigenvector CENTRALITY”
L175: Suggest changing “the airline transportation network” to “AN airline transportation network”
L184-187: I am not a huge fan of this section as it implies the choice of measure should depend predominantly on network structure. While, this is undoubtedly important the primary/over-riding factor needs to be the biological question being addressed. Would suggest at least mentioning this here.
L297: Suggest changing “macaque monkeys” to simply read “macaques”
L410&428: Is this computer code available anywhere that could be linked to after these statements (supplementary, GitHub etc)?
L537-542: I feel these few sentences could do with some careful revision as they are rather confusing at the moment. In part this is because rank might refer both to dominance rank and centrality rank – so finding a way to avoid this confusion would help a lot. In L540 would suggest “…but THIS was NOT STATISTICALLY significant according…”
L575: Suggest “only reached STATISTICAL significance”
L587-588: Little bit unclear and would suggest being more explicit and saying most central or one of the most central rather than simply high social centrality to make it as clear as possible you are talking about the top one (or few) individual(s)
Fig. 1 legend: Would it help to say “…is the FIRST one which falls below the threshold…”
Fig. 3 legend: Would be helpful to describe what the branching at the top depicts
Would be good to make the figures suitable for printing in B&W given the simple colour schemes (currently they are not!) – obviously this is not essential though
For Tables 3-6 my preference is to report the quantiles themselves as it is these that are important for comparisons to the observed parameters – I can understand if they don’t fit in the main text table but might good to provide in Supplementary if possible?

I hope my comments are useful in improving the paper,
Matthew Silk

---

## Round 0.3 · accepted · Accept

The authors have adequately addressed the remaining comments.